# A Fully Decentralized Surrogate for Multi-Agent Policy Optimization

**Kefan Su**                                              *sukefan@pku.edu.cn*
*School of Computer Science*
*Peking University*

**Zongqing Lu**[†]                                        *zongqing.lu@pku.edu.cn*
*School of Computer Science*
*Peking University*

**Reviewed on OpenReview:** *https://openreview.net/forum?id=MppUW9OuU2*

## Abstract

The study of fully decentralized learning or independent learning in cooperative multi-agent reinforcement learning has a history of decades. Recent empirical studies have shown that independent PPO (IPPO) can achieve good performance, comparable to or even better than the methods of centralized training with decentralized execution, in several benchmarks. However, a decentralized actor-critic algorithm with a convergence guarantee is still an open problem. In this paper, we propose *decentralized policy optimization* (DPO), a decentralized actor-critic algorithm with monotonic improvement and convergence guarantee. We derive a novel decentralized surrogate for policy optimization such that the monotonic improvement of joint policy can be guaranteed by each agent *independently* optimizing the surrogate. For practical implementation, this decentralized surrogate can be realized by two adaptive coefficients for policy optimization at each agent. Empirically, we evaluate DPO, IPPO, and independent Q-learning (IQL) in a variety of cooperative multi-agent tasks, covering discrete and continuous action spaces, as well as fully and partially observable environments. The results show DPO outperforms both IPPO and IQL in most tasks, which serves as evidence for our theoretical results. The code is available at `https://github.com/PKU-RL/DPO`.

## 1 Introduction

In cooperative multi-agent reinforcement learning (MARL), centralized training with decentralized execution (CTDE) has been the dominant framework (Lowe et al., 2017; Foerster et al., 2018; Sunehag et al., 2018; Rashid et al., 2018; Wang et al., 2021a; Zhang et al., 2021; Yu et al., 2021). Such a framework resolves the non-stationarity problem with the centralized value function that takes the global information as input, making it beneficial to the training process. Conversely, decentralized learning has received less attention. The main reason is likely due to the fact that few theoretical guarantees exist for decentralized learning and the interpretability is insufficient even though the simplest form of decentralized learning, *i.e.*, independent learning, may achieve good empirical performance in several benchmarks (Papoudakis et al., 2021). However, decentralized learning itself still should be considered as there are still many settings in which the global information is unavailable, and also for better robustness and scalability (Zhang et al., 2019). Moreover, decentralized learning is straightforward, comprehensible, and easy to implement in practice.

Independent Q-learning (IQL) (Tampuu et al., 2015) and independent PPO (IPPO) (de Witt et al., 2020) are the straightforward decentralized learning methods for cooperative MARL, where each agent learns the policy by DQN (Mnih et al., 2015) and PPO (Schulman et al., 2017) respectively. Empirical studies (de Witt et al., 2020; Yu et al., 2021; Papoudakis et al., 2021) demonstrate that these two methods can obtain good

---

[†]Corresponding Author

performance, close to CTDE methods. Especially, IPPO can outperform several CTDE methods in a few benchmarks, including MPE (Lowe et al., 2017) and SMAC (Samvelyan et al., 2019), which shows great promise for decentralized learning. Unfortunately, to the best of our knowledge, there is still no theoretical guarantee or rigorous explanation for IPPO, though there has been some study (Sun et al., 2022).

In this paper, we take a step further and propose *decentralized policy optimization* (**DPO**), a fully decentralized actor-critic method with monotonic improvement and convergence guarantee for cooperative MARL. Similar to IPPO, DPO is actually *independent learning* as each agent optimizes its own objective individually and independently in DPO. However, unlike IPPO, such an independent policy optimization of DPO can guarantee the monotonic improvement of the joint policy.

From the essence of fully decentralized learning, we first analyze the Q-function in the decentralized setting and further show that the optimization objective of IPPO may not induce joint policy improvement. Then, starting from the joint monotonic objective from the theoretical results in single-agent RL (Grudzien et al., 2022; Lan, 2023) and together considering the characteristics of fully decentralized learning, we introduce a *novel* lower bound of joint policy improvement as the surrogate for decentralized policy optimization. This surrogate can be naturally decomposed for each agent, which means each agent can optimize its individual objective to make sure that the joint policy improves monotonically. Practically, this decentralized surrogate can be realized by two adaptive coefficients for policy optimization at each agent. The idea of DPO is simple yet effective and well-suited for fully decentralized learning.

Empirically, we evaluate the performance of DPO, IPPO, and IQL in a variety of cooperative multi-agent tasks, including a cooperative stochastic game, MPE (Lowe et al., 2017), multi-agent MuJoCo (Peng et al., 2021), and SMAC (Samvelyan et al., 2019). Our evaluation covers discrete and continuous action spaces, as well as fully and partially observable environments. The results indicated that DPO outperforms both IPPO and IQL in most tasks, which serves as evidence for our theoretical results.

## 2 Related Work

**CTDE.** In cooperative MARL, centralized training with decentralized execution (CTDE) is the most popular framework (Lowe et al., 2017; Iqbal & Sha, 2019; Foerster et al., 2018; Sunehag et al., 2018; Rashid et al., 2018; Wang et al., 2021a; Zhang et al., 2021; Peng et al., 2021). CTDE algorithms address the non-stationarity problem in the multi-agent environment by the centralized value function. One line of research in CTDE is value decomposition (Sunehag et al., 2018; Rashid et al., 2018; Son et al., 2019; Yang et al., 2020; Wang et al., 2021a), where a joint Q-function is learned and factorized into local Q-functions by the relationship between optimal joint action and optimal local actions. Another line of research in CTDE is multi-agent actor-critic (Foerster et al., 2018; Iqbal & Sha, 2019; Wang et al., 2021b; Zhang et al., 2021; Su & Lu, 2022; Wang et al., 2023a), where the centralized value function is learned to provide policy gradients for agents to learn stochastic policies. More recently, policy optimization has attracted much attention for cooperative MARL. PPO (Schulman et al., 2017) and TRPO (Schulman et al., 2015a) have been extended to multi-agent settings by MAPPO (Yu et al., 2021), CoPPO (Wu et al., 2021), HAPPO (Kuba et al., 2021) and A2PO (Wang et al., 2023b) respectively via learning a centralized state value function. However, these methods are CTDE and thus not appropriate for decentralized learning.

**Fully decentralized learning.** Independent learning (OroojlooyJadid & Hajinezhad, 2019) is the most straightforward approach for fully decentralized learning and has been a subject of study in cooperative MARL for decades. The representatives are independent Q-learning (IQL) (Tan, 1993; Tampuu et al., 2015) and independent actor-critic (IAC) as Foerster et al. (2018) empirically studied. These methods enable agents to directly execute the single-agent Q-learning or actor-critic algorithm individually. The drawback of such independent learning methods is obvious. As other agents are also learning, each agent interacts with a non-stationary environment, which violates the stationary condition of Markov decision processes (MDPs). Thus, these methods are not with any convergence guarantee theoretically, though IQL could obtain good performance in several benchmarks (Papoudakis et al., 2021). More recently, decentralized learning has also been specifically studied with communication (Zhang et al., 2018; Li et al., 2020) or parameter sharing (Terry et al., 2020). However, in this paper, we consider fully decentralized learning in the strictest sense – with each agent independently learning its policy while being not allowed to communicate or share parameters as

in Tampuu et al. (2015); de Witt et al. (2020). We will propose an algorithm with convergence guarantees in such a fully decentralized learning setting.

**IPPO.** TRPO (Schulman et al., 2015a) is an important single-agent actor-critic algorithm that limits the policy update in a trust region and ensures monotonic improvement by optimizing a surrogate objective. PPO (Schulman et al., 2017) is a practical but effective algorithm derived from TRPO. PPO replaces the trust region constraint with a simpler clip trick. IPPO (de Witt et al., 2020) is a recently emerged cooperative MARL algorithm in which each agent just learns with independent PPO. Though IPPO is still with no convergence guarantee, it obtains surprisingly good performance in SMAC (Samvelyan et al., 2019). IPPO is further empirically studied by Yu et al. (2021); Papoudakis et al. (2021). Their results show IPPO can outperform a few CTDE methods in several benchmark tasks. These studies highlight the potential of policy optimization in fully decentralized learning, a topic on which this paper focuses.

## 3 Decentralized Policy Optimization

From the perspective of policy optimization, in fully decentralized learning, we need to find an objective for each agent such that joint policy improvement can be guaranteed by each agent independently and individually optimizing its own objective. Therefore, we propose a novel lower bound of the joint policy improvement to enable *decentralized policy optimization* (DPO). In the following, we first discuss some preliminaries; then we analyze the critic in fully decentralized learning; next, we derive the decentralized surrogate and prove the convergence; finally, we introduce the practical algorithm of DPO.

### 3.1 Preliminaries

**Dec-POMDP.** Decentralized partially observable Markov decision process is a general model for cooperative MARL. A Dec-POMDP is a tuple $\mathcal{G} = \{S, A, P, Y, O, I, N, r, \gamma, \mu\}$. $S$ is the state space, $N$ is the number of agents, $\gamma \in [0, 1)$ is the discount factor, and $I = \{1, 2 \cdots N\}$ is the set of all agents. $A = A_1 \times A_2 \times \cdots \times A_N$ represents the joint action space, where $A_i$ is the individual action space for agent $i$. $P(s'|s, \boldsymbol{a}) : S \times A \times S \rightarrow [0, 1]$ is the transition function, and $r(s, \boldsymbol{a}) : S \times A \rightarrow [-r_{\max}, r_{\max}]$ is the reward function of state $s \in S$ and joint action $\boldsymbol{a} \in A$, where $r_{\max}$ is bound of the reward function. $Y$ is the observation space, and $O(s, i) : S \times I \rightarrow Y$ is a mapping from state to observation for each agent $i$. The objective of Dec-POMDP is to maximize $J(\boldsymbol{\pi}) = \mathbb{E}_{\boldsymbol{\pi}} \left[ \sum_{t=0} \gamma^t r(s_t, \boldsymbol{a}_t) \right]$, thus we need to find the optimal joint policy $\boldsymbol{\pi}^* = \arg\max_{\boldsymbol{\pi}} J(\boldsymbol{\pi})$. To settle the partial observable problem, history $\tau_i \in \mathcal{T}_i = (Y \times A_i)^*$ is often used to replace observation $o_i \in Y$. In fully decentralized learning, each agent $i$ independently learns an individual policy $\pi^i(a_i|\tau_i)$ and their joint policy $\boldsymbol{\pi}$ can be represented as the product of each $\pi^i$. Though each agent learns individual policy as $\pi^i(a_i|\tau_i)$ in practice, in our analysis, we will assume that each agent could receive the state $s$ because the analysis in partially observable environments is much more difficult and the problem may be undecidable in Dec-POMDP (Madani et al., 1999). Moreover, the V-function and Q-function of the joint policy $\boldsymbol{\pi}$ are as follows,

$$V^{\boldsymbol{\pi}}(s) = \mathbb{E}_{\boldsymbol{a} \sim \boldsymbol{\pi}} \left[ Q^{\boldsymbol{\pi}}(s, \boldsymbol{a}) \right] \tag{1}$$

$$Q^{\boldsymbol{\pi}}(s, \boldsymbol{a}) = r(s, \boldsymbol{a}) + \gamma \mathbb{E}_{s' \sim P(\cdot|s, \boldsymbol{a})} \left[ V^{\boldsymbol{\pi}}(s') \right]. \tag{2}$$

In our discussion, we assume the initial state $s_0$ is sampled from a fixed distribution $\mu$. The objective $J(\boldsymbol{\pi})$ can be rewritten as $J(\boldsymbol{\pi}) = \mathbb{E}_{s_0 \sim \mu}[V^{\boldsymbol{\pi}}(s_0)]$. We will use the value function without the agent index to represent the joint value function related to the joint action such as $Q^{\boldsymbol{\pi}}(s, \boldsymbol{a})$ and the value function with the agent index $i$ to represent the individual value function of the agent $i$ such as $Q^{\pi^i}_{\pi^{-i}}(s, a_i)$ which will be discussed later.

**Joint Objective for Monotonic Improvement.** In Dec-POMDP, we can still obtain an objective for the joint policy $\boldsymbol{\pi}$ from the theoretical results in single-agent RL (Grudzien et al., 2022; Lan, 2023) to ensure the joint policy can improve monotonically.

**Lemma 3.1.** *Suppose $\boldsymbol{\pi}_{\text{new}}$ and $\boldsymbol{\pi}_{\text{old}}$ are two joint policies. If $\boldsymbol{\pi}_{\text{new}}$ and $\boldsymbol{\pi}_{\text{old}}$ satisfy the condition*

$$\mathcal{L}^{\text{joint}}_{\boldsymbol{\pi}_{\text{old}}}(\boldsymbol{\pi}_{\text{new}}, s) - C \cdot D_{\text{KL}}(\boldsymbol{\pi}_{\text{old}}(\cdot|s) \| \boldsymbol{\pi}_{\text{new}}(\cdot|s)) \geq 0, \ \forall s \in S, \tag{3}$$

*where $\mathcal{L}_{\boldsymbol{\pi}_{\mathrm{old}}}^{\mathrm{joint}}(\boldsymbol{\pi}, s) = \sum_{\boldsymbol{a}} \boldsymbol{\pi}(\boldsymbol{a}|s) A_{\mathrm{old}}(s, \boldsymbol{a})$, $A_{\mathrm{old}}$ is the advantage function under $\boldsymbol{\pi}_{\mathrm{old}}$, and $C$ is a constant, then $V^{\boldsymbol{\pi}_{\mathrm{new}}}(s) \geq V^{\boldsymbol{\pi}_{\mathrm{old}}}(s)$.*

This Lemma is the corollary of Lemma 3.3 in Grudzien et al. (2022).

From Lemma 3.1, we can define an objective as $\mathcal{L}_{\boldsymbol{\pi}_{\mathrm{old}}}^{\mathrm{joint}}(\boldsymbol{\pi}, s) - C \cdot D_{\mathrm{KL}}(\boldsymbol{\pi}_{\mathrm{old}}(\cdot|s) \| \boldsymbol{\pi}(\cdot|s))$, which is referred as joint monotonic objective. Maximizing this objective can guarantee that the joint policy is improving monotonically. However, the joint monotonic objective cannot be directly optimized in fully decentralized learning as this objective is involved in the joint policy, which cannot be accessed in fully decentralized settings.

We will propose a new lower bound (surrogate) for the joint monotonic objective, which can be optimized in fully decentralized learning. Before introducing our new surrogate, we need to first analyze the critic of the agent in fully decentralized learning, which is referred to as decentralized critic.

## 3.2 Decentralized Critic

In fully decentralized learning, each agent learns independently from its own interactions with the environment. Therefore, the Q-function of each agent $i$ is the following formula:

$$Q_{\pi^{-i}}^{\pi^i}(s, a_i) = r_{\pi^{-i}}(s, a_i) + \gamma \mathbb{E}_{a_{-i} \sim \pi^{-i}, s' \sim P(\cdot|s, a_i, a_{-i}), a_i' \sim \pi^i}[Q_{\pi^{-i}}^{\pi^i}(s', a_i')], \tag{4}$$

where $r_{\pi^{-i}}(s, a_i) = \mathbb{E}_{\pi^{-i}}[r(s, a_i, a_{-i})]$, and $\pi^{-i}$ and $a_{-i}$ respectively denote the joint policy and joint action of all agents expect agent $i$. If we take the expectation $\mathbb{E}_{a_{-i}' \sim \pi^{-i}(\cdot|s'), a_{-i} \sim \pi^{-i}(\cdot|s)}$ over both sides of the Q-function of joint policy (2), then we have

$$\mathbb{E}_{\pi^{-i}}[Q^{\boldsymbol{\pi}}(s, a_i, a_{-i})] = r_{\pi^{-i}}(s, a_i) + \gamma \mathbb{E}_{a_{-i} \sim \pi^{-i}, s' \sim P(\cdot|s, a_i, a_{-i}), a_i' \sim \pi^i} \left[ \mathbb{E}_{\pi^{-i}}[Q^{\boldsymbol{\pi}}(s', a_i', a_{-i}')] \right].$$

We can see that $Q_{\pi^{-i}}^{\pi^i}(s, a_i)$ and $\mathbb{E}_{\pi^{-i}}[Q^{\boldsymbol{\pi}}(s, a_i, a_{-i})]$ satisfy the same iteration. Moreover, we will show in the following that $Q_{\pi^{-i}}^{\pi^i}(s, a_i)$ and $\mathbb{E}_{\pi^{-i}}[Q^{\boldsymbol{\pi}}(s, a_i, a_{-i})]$ are just the same.

We first define an operator $\Gamma_{\pi^{-i}}^{\pi^i}$ as follows,

$$\Gamma_{\pi^{-i}}^{\pi^i} Q(s, a_i) = r_{\pi^{-i}}(s, a_i) + \gamma \mathbb{E}_{a_{-i} \sim \pi^{-i}, s' \sim P(\cdot|s, a_i, a_{-i}), a_i' \sim \pi^i}[Q(s', a_i')]. \tag{5}$$

Then we will prove that the operator $\Gamma_{\pi^{-i}}^{\pi^i}$ is a contraction. Considering any two individual Q-functions $Q_1$ and $Q_2$, we have:

$$\begin{aligned}
\|\Gamma_{\pi^{-i}}^{\pi^i} Q_1 - \Gamma_{\pi^{-i}}^{\pi^i} Q_2\|_\infty &= \max_{s, a_i} \gamma |\mathbb{E}_{a_{-i}, s', a_i'}[Q_1(s', a_i') - Q_2(s', a_i')]| \\
&\leq \gamma \mathbb{E}_{a_{-i}, s', a_i'}[\max_{s', a_i'} |Q_1(s', a_i') - Q_2(s', a_i')|] = \gamma \max_{s', a_i'} |Q_1(s', a_i') - Q_2(s', a_i')| \\
&= \gamma \|Q_1 - Q_2\|_\infty. \tag{6}
\end{aligned}$$

So the operator $\Gamma_{\pi^{-i}}^{\pi^i}$ has one and only one fixed point, which means

$$Q_{\pi^{-i}}^{\pi^i}(s, a_i) = \mathbb{E}_{\pi^{-i}}[Q^{\boldsymbol{\pi}}(s, a_i, a_{-i})], V_{\pi^{-i}}^{\pi^i}(s) = \mathbb{E}_{\pi^{-i}}[V^{\boldsymbol{\pi}}(s)] = V^{\boldsymbol{\pi}}(s).$$

This conclusion is from the perspective of definition and policy evaluation. Further discussion about the learning or iteration of $Q_{\pi^{-i}}^{\pi^i}(s, a_i)$ is included in Appendix E.

With this well-defined decentralized critic, we can further analyze the objective of IPPO (de Witt et al., 2020). In IPPO, the policy objective (without clipping) of each agent $i$ in state $s$ can be essentially formulated as follows:

$$\mathcal{L}_{\boldsymbol{\pi}_{\mathrm{old}}}^i(\pi^i, s) = \sum_{a_i} \pi^i(a_i|s) A_{\mathrm{old}}^i(s, a_i), \tag{7}$$

where $A_{\mathrm{old}}^i(s, a_i) = Q_{\pi_{\mathrm{old}}^{-i}}^{\pi_{\mathrm{old}}^i}(s, a_i) - \mathbb{E}_{\pi_{\mathrm{old}}^i}[Q_{\pi_{\mathrm{old}}^{-i}}^{\pi_{\mathrm{old}}^i}(s, a_i)] = \mathbb{E}_{\pi_{\mathrm{old}}^{-i}}[A_{\mathrm{old}}(s, a_i, a_{-i})].$

However, (7) is different from (3) in the joint monotonic objective. Thus, directly optimizing (7) may not improve the joint policy, and thus cannot provide any guarantee for convergence, to the best of our knowledge. Nevertheless, it seems that $A_{\text{old}}^i(s, a_i)$ is the only advantage formulation that can be accessed by each agent in fully decentralized learning. So, the policy objective of DPO will be derived on (7) but with modifications to guarantee convergence, and we will introduce the details in the next section. In the following, we discuss how to compute this advantage in practice in fully decentralized learning.

As we need to calculate $A_{\text{old}}^i(s, a_i) = \mathbb{E}_{\pi_{\text{old}}^{-i}}[r(s, a_i, a_{-i}) + \gamma V^{\boldsymbol{\pi}_{\text{old}}}(s') - V^{\boldsymbol{\pi}_{\text{old}}}(s)]$ for the policy update, we can approximate $A_{\text{old}}^i(s, a_i)$ with $\hat{A}^i(s, a_i) = r + \gamma V_{\pi^{-i}}^{\pi^i}(s') - V_{\pi^{-i}}^{\pi^i}(s)$, which is an unbiased estimate of $A_{\text{old}}^i(s, a_i)$, though it may be with a large variance. In practice, we can follow the traditional idea in fully decentralized learning, and let each agent $i$ independently learn an individual value function $V^i(s)$. Then, we further have $\hat{A}^i(s, a_i) \approx r + \gamma V^i(s') - V^i(s)$. The loss for the decentralized critic is as follows:

$$\mathcal{L}_{\text{critic}}^i = \mathbb{E}\left[(V^i(s) - y_i)^2\right], \text{ where } y_i = r + \gamma V^i(s') \text{ or other target values.} \tag{8}$$

Here we could take the target value $y_i$ according to different methods like Monte Carlo returns or GAE (Schulman et al., 2015b). There may be some ways to improve the learning of this critic, which however is beyond the scope of this paper.

### 3.3 Decentralized Surrogate

We are ready to introduce the decentralized surrogate. First we will discuss the relationship between the joint policy objective $\mathcal{L}_{\boldsymbol{\pi}_{\text{old}}}^{\text{joint}}(\boldsymbol{\pi}, s)$ and the individual policy objective $\mathcal{L}_{\boldsymbol{\pi}_{\text{old}}}^i(\pi^i, s)$. We have the following lemma.

**Lemma 3.2.** *Suppose $\boldsymbol{\pi}_{\text{old}}$ and $\boldsymbol{\pi}$ are two joint policies. Then, the following bound holds for any agent $i$:*

$$\mathcal{L}_{\boldsymbol{\pi}_{\text{old}}}^{\text{joint}}(\boldsymbol{\pi}, s) - \mathcal{L}_{\boldsymbol{\pi}_{\text{old}}}^i(\pi^i, s) \geq -\tilde{M}\sqrt{\sum_{j \neq i} D_{\text{KL}}(\pi_{\text{old}}^j(\cdot|s)\|\pi^j(\cdot|s))}, \text{ where } \tilde{M} = \frac{2r_{\max}}{1 - \gamma}. \tag{9}$$

*Proof.* We first consider $\mathcal{L}_{\boldsymbol{\pi}_{\text{old}}}^{\text{joint}}(\boldsymbol{\pi}, s) - \mathcal{L}_{\boldsymbol{\pi}_{\text{old}}}^i(\pi^i, s)$. According to (3) and (7), we have the following equation:

$$\mathcal{L}_{\boldsymbol{\pi}_{\text{old}}}^{\text{joint}}(\boldsymbol{\pi}, s) - \mathcal{L}_{\boldsymbol{\pi}_{\text{old}}}^i(\pi^i, s) = \mathbb{E}_{\pi^i}\left[\sum_{a_{-i}}\left(\pi^{-i}(a_{-i}|s) - \pi_{\text{old}}^{-i}(a_{-i}|s)\right)A_{\text{old}}(s, a_i, a_{-i})\right].$$

Then, we have the following inequalities:

$$|\mathcal{L}_{\boldsymbol{\pi}_{\text{old}}}^{\text{joint}}(\boldsymbol{\pi}, s) - \mathcal{L}_{\boldsymbol{\pi}_{\text{old}}}^i(\pi^i, s)| \leq \mathbb{E}_{\pi^i}\left[\sum_{a_{-i}}|\pi_{\text{old}}^{-i}(a_{-i}|s) - \pi^{-i}(a_{-i}|s)||A_{\text{old}}(s, a_i, a_{-i})|\right]$$

$$\leq \mathbb{E}_{\pi^i}\left[M\sum_{a_{-i}}|\pi_{\text{old}}^{-i}(a_{-i}|s) - \pi^{-i}(a_{-i}|s)|\right] \quad (M = \frac{r_{\max}}{1 - \gamma} \geq \max_{s, \boldsymbol{a}}|A_{\text{old}}(s, \boldsymbol{a})|)$$

$$= 2M D_{\text{TV}}(\pi_{\text{old}}^{-i}(\cdot|s)\|\pi^{-i}(\cdot|s))$$

$$= \tilde{M} D_{\text{TV}}(\pi_{\text{old}}^{-i}(\cdot|s)\|\pi^{-i}(\cdot|s)) \quad (\tilde{M} = 2M)$$

$$\leq \tilde{M}\sqrt{D_{\text{KL}}(\pi_{\text{old}}^{-i}(\cdot|s)\|\pi^{-i}(\cdot|s))} \tag{10}$$

$$= \tilde{M}\sqrt{\sum_{j \neq i} D_{\text{KL}}(\pi_{\text{old}}^j(\cdot|s)\|\pi^j(\cdot|s))}, \tag{11}$$

where (10) is from the relationship between the total variation distance and KL-divergence that $D_{\text{TV}}(p\|q)^2 \leq D_{\text{KL}}(p\|q)$ (Schulman et al., 2015a), and (11) is a property of the KL-divergence that $D_{\text{KL}}(\boldsymbol{\pi}_{\text{old}}(\cdot|s)\|\boldsymbol{\pi}(\cdot|s)) = \sum_i D_{\text{KL}}(\pi_{\text{old}}^i(\cdot|s)\|\pi^i(\cdot|s))$. From (11), we can further obtain the following inequality, which completes the proof,

$$\mathcal{L}_{\boldsymbol{\pi}_{\text{old}}}^{\text{joint}}(\boldsymbol{\pi}, s) - \mathcal{L}_{\boldsymbol{\pi}_{\text{old}}}^i(\pi^i, s) \geq -\tilde{M}\sqrt{\sum_{j \neq i} D_{\text{KL}}(\pi_{\text{old}}^j(\cdot|s)\|\pi^j(\cdot|s))}.$$

$\square$

We need to emphasize that the inequality (9) connects the joint policy objective with the individual policy objective, which is essential for our purpose of fully decentralized learning.

Next, we will derive our novel lower bound of joint policy improvement by the following theorem.

**Theorem 3.3.** *Suppose $\boldsymbol{\pi}_{\text{old}}$ and $\boldsymbol{\pi}$ are two joint policies. We have*

$$\mathcal{L}_{\boldsymbol{\pi}_{\text{old}}}^{\text{joint}}(\boldsymbol{\pi}, s) - C \cdot D_{\text{KL}}(\boldsymbol{\pi}_{\text{old}}(\cdot|s)\|\boldsymbol{\pi}(\cdot|s)) \geq \frac{1}{N} \sum_{i=1}^{N} \mathcal{L}_{\boldsymbol{\pi}_{\text{old}}}^{i}(\pi^i, s) - \hat{M} \sum_{i=1}^{N} \sqrt{D_{\text{KL}}(\pi_{\text{old}}^i(\cdot|s)\|\pi^i(\cdot|s))}$$

$$- C \sum_{i=1}^{N} D_{\text{KL}}(\pi_{\text{old}}^i(\cdot|s)\|\pi^i(\cdot|s)) \geq 0, \ \forall s \in S, \qquad (12)$$

*where $\hat{M} = \sqrt{\frac{N-1}{N}} \frac{2r_{\max}}{1-\gamma}$ and $C > 0$ is any constant.*

*Proof.* We will start to prove this theorem from the objective $\mathcal{L}_{\boldsymbol{\pi}_{\text{old}}}^{\text{joint}}(\boldsymbol{\pi}, s) - C \cdot D_{\text{KL}}(\boldsymbol{\pi}_{\text{old}}(\cdot|s)\|\boldsymbol{\pi}(\cdot|s))$,

$$\mathcal{L}_{\boldsymbol{\pi}_{\text{old}}}^{\text{joint}}(\boldsymbol{\pi}, s) - C \cdot D_{\text{KL}}(\boldsymbol{\pi}_{\text{old}}(\cdot|s)\|\boldsymbol{\pi}(\cdot|s)) = \frac{1}{N} \sum_{i=1}^{N} \mathcal{L}_{\boldsymbol{\pi}_{\text{old}}}^{\text{joint}}(\boldsymbol{\pi}, s) - C \cdot D_{\text{KL}}(\boldsymbol{\pi}_{\text{old}}(\cdot|s)\|\boldsymbol{\pi}(\cdot|s))$$

$$\geq \frac{1}{N} \sum_{i=1}^{N} \mathcal{L}_{\boldsymbol{\pi}_{\text{old}}}^{i}(\pi^i, s) - \frac{\tilde{M}}{N} \sum_{i=1}^{N} \sqrt{\sum_{j\neq i} D_{\text{KL}}(\pi_{\text{old}}^j(\cdot|s)\|\pi^j(\cdot|s))} - C \cdot D_{\text{KL}}(\boldsymbol{\pi}_{\text{old}}(\cdot|s)\|\boldsymbol{\pi}(\cdot|s)) \qquad (13)$$

$$\geq \frac{1}{N} \sum_{i=1}^{N} \mathcal{L}_{\boldsymbol{\pi}_{\text{old}}}^{i}(\pi^i, s) - \tilde{M} \sqrt{\frac{N-1}{N} \sum_{i=1}^{N} D_{\text{KL}}(\pi_{\text{old}}^i(\cdot|s)\|\pi^i(\cdot|s))} - C \cdot D_{\text{KL}}(\boldsymbol{\pi}_{\text{old}}(\cdot|s)\|\boldsymbol{\pi}(\cdot|s)) \qquad (14)$$

$$= \frac{1}{N} \sum_{i=1}^{N} \mathcal{L}_{\boldsymbol{\pi}_{\text{old}}}^{i}(\pi^i, s) - \tilde{M} \sqrt{\frac{N-1}{N} \sum_{i=1}^{N} D_{\text{KL}}(\pi_{\text{old}}^i(\cdot|s)\|\pi^i(\cdot|s))} - C \sum_{i=1}^{N} D_{\text{KL}}(\pi_{\text{old}}^i(\cdot|s)\|\pi^i(\cdot|s)) \qquad (15)$$

$$\geq \frac{1}{N} \sum_{i=1}^{N} \mathcal{L}_{\boldsymbol{\pi}_{\text{old}}}^{i}(\pi^i, s) - \hat{M} \sum_{i=1}^{N} \sqrt{D_{\text{KL}}(\pi_{\text{old}}^i(\cdot|s)\|\pi^i(\cdot|s))} - C \sum_{i=1}^{N} D_{\text{KL}}(\pi_{\text{old}}^i(\cdot|s)\|\pi^i(\cdot|s)). \qquad (16)$$

The inequality (13) is the direct application of the inequality (9) in Lemma 3.2. The inequality (14) is from the Cauchy-Schwarz inequality,

$$\sum_{i=1}^{N} \sqrt{\sum_{j\neq i} D_{\text{KL}}(\pi_{\text{old}}^j(\cdot|s)\|\pi^j(\cdot|s))} \leq \sqrt{N \sum_{i=1}^{N} \sum_{j\neq i} D_{\text{KL}}(\pi_{\text{old}}^j(\cdot|s)\|\pi^j(\cdot|s))}$$

$$= \sqrt{N(N-1) \sum_{i=1}^{N} D_{\text{KL}}(\pi_{\text{old}}^i(\cdot|s)\|\pi^i(\cdot|s))}.$$

The inequality (15) is from a property of the KL-divergence (see the proof of Lemma 3.2), while the inequality (16) is from the simple inequality $\sqrt{\sum_i a_i} \leq \sum_i \sqrt{a_i}$ ($a_i \geq 0, \forall i$). $\qquad\square$

The lower bound in Theorem 3.3 is dedicated to decentralized policy optimization because it can be directly decomposed individually for each agent as a decentralized surrogate. From Theorem 3.3, if we set the policy optimization objective of each agent $i$ as

$$\pi_{\text{new}}^i(\cdot|s) = \arg\max_{\pi^i} \left( \frac{1}{N} \mathcal{L}_{\boldsymbol{\pi}_{\text{old}}}^{i}(\pi^i, s) - \hat{M} \sqrt{D_{\text{KL}}(\pi_{\text{old}}^i(\cdot|s)\|\pi^i(\cdot|s))} - C \cdot D_{\text{KL}}(\pi_{\text{old}}^i(\cdot|s)\|\pi^i(\cdot|s)) \right), \qquad (17)$$

then we have $J(\boldsymbol{\pi}_{\text{new}}) \geq J(\boldsymbol{\pi}_{\text{old}})$ from Lemma 3.1. Finally, we can obtain the following theorem.

**Theorem 3.4.** *If we define a joint policy sequence $\{\boldsymbol{\pi}_t\}$ as follows:*

$$\pi_{t+1}^i(\cdot|s) = \arg\max_{\pi^i} \left( \frac{1}{N} \mathcal{L}_{\boldsymbol{\pi}_t}^i(\pi^i, s) - \hat{M}\sqrt{D_{\mathrm{KL}}(\pi_t^i(\cdot|s)\|\pi^i(\cdot|s))} - C \cdot D_{\mathrm{KL}}(\pi_t^i(\cdot|s)\|\pi^i(\cdot|s)) \right) \forall i \in I, \forall s \in S,$$
(18)

*then the sequence $\{J(\boldsymbol{\pi}_t)\}$ will improve monotonically and converge to sub-optimum.*

*Proof.* At each iteration, as the policy of each agent is obtained by (18), all agents jointly maximize the RHS of (12). Thus, from Theorem 3.3, we have $\forall s \in S$

$$\mathcal{L}_{\boldsymbol{\pi}_t}^{\mathrm{joint}}(\boldsymbol{\pi}_{t+1}, s) - C \cdot D_{\mathrm{KL}}(\boldsymbol{\pi}_t(\cdot|s)\|\boldsymbol{\pi}_{t+1}(\cdot|s))$$

$$\geq \frac{1}{N}\sum_{i=1}^{N}\mathcal{L}_{\boldsymbol{\pi}_t}^i(\pi_{t+1}^i, s) - \hat{M}\sum_{i=1}^{N}\sqrt{D_{\mathrm{KL}}(\pi_t^i(\cdot|s)\|\pi_{t+1}^i(\cdot|s))} - C\sum_{i=1}^{N}D_{\mathrm{KL}}(\pi_t^i(\cdot|s)\|\pi_{t+1}^i(\cdot|s))$$

$$\geq \frac{1}{N}\sum_{i=1}^{N}\mathcal{L}_{\boldsymbol{\pi}_t}^i(\pi_t^i, s) - \hat{M}\sum_{i=1}^{N}\sqrt{D_{\mathrm{KL}}(\pi_t^i(\cdot|s)\|\pi_t^i(\cdot|s))} - C\sum_{i=1}^{N}D_{\mathrm{KL}}(\pi_t^i(\cdot|s)\|\pi_t^i(\cdot|s)) = 0, \quad (19)$$

where (19) is from the definition of $\pi_{t+1}^i(\cdot|s)$ in (18). From Lemma 3.1, we know $V^{\boldsymbol{\pi}_{t+1}}(s) \geq V^{\boldsymbol{\pi}_t}(s)$, $\forall s \in S$. From the definition $J(\boldsymbol{\pi}) = \mathbb{E}_{s_0\sim\mu}[V^{\boldsymbol{\pi}}(s_0)]$, we know $J(\boldsymbol{\pi}_{t+1}) \geq J(\boldsymbol{\pi}_t)$, which means that the sequence $\{J(\boldsymbol{\pi}_t)\}$ improves monotonically. Moreover, as $\{J(\boldsymbol{\pi}_t)\}$ is bounded (the reward function is bounded), $\{J(\boldsymbol{\pi}_t)\}$ will converge to sub-optimum, which completes the proof. $\square$

As (17) can be independently optimized at each agent, the monotonic improvement and convergence of joint policy can be achieved by *fully decentralized policy optimization*. It is worth noting that the result above is under the assumption that each agent can obtain the state, and in practice, each agent can take the individual trajectory $\tau_i$ as the approximation to the state.

## 3.4 Remarks About Our Theoretical Results

In this section, we would like to summarize our theoretical results and discuss an important problem: how could DPO overcome the non-stationarity problem in decentralized learning and obtain the convergence guarantee?

Our general idea is that finding a surrogate for the joint monotonic improvement condition $\mathcal{L}_{\boldsymbol{\pi}_{\mathrm{old}}}^{\mathrm{joint}}(\boldsymbol{\pi}_{\mathrm{new}}, s) - C \cdot D_{\mathrm{KL}}(\boldsymbol{\pi}_{\mathrm{old}}(\cdot|s)\|\boldsymbol{\pi}_{\mathrm{new}}(\cdot|s))$ and this surrogate can be optimized independently for each agent. Then optimizing this surrogate will make sure that the joint policy will improve monotonically.

DPO realizes this idea in two stages. In the first stage, according to the inequality (9) in Lemma 3.2, we have a bound between $\mathcal{L}_{\boldsymbol{\pi}_{\mathrm{old}}}^{\mathrm{joint}}(\boldsymbol{\pi}, s)$ and $\mathcal{L}_{\boldsymbol{\pi}_{\mathrm{old}}}^i(\pi^i, s)$. This bound actually help us to replace $\mathbb{E}_{\pi_{\mathrm{new}}^{-i}}[A_{\mathrm{old}}(s, a_i, a_{-i})]$ in $\mathcal{L}_{\boldsymbol{\pi}_{\mathrm{old}}}^{\mathrm{joint}}(\boldsymbol{\pi}, s)$ with $A_{\mathrm{old}}^i(s, a_i) = \mathbb{E}_{\pi_{\mathrm{old}}^{-i}}[A_{\mathrm{old}}(s, a_i, a_{-i})]$, where $\pi_{\mathrm{new}}^{-i}$ is not accessible in training. But the extra terms $\tilde{M}\sqrt{\sum_{j\neq i}D_{\mathrm{KL}}(\pi_{\mathrm{old}}^j(\cdot|s)\|\pi^j(\cdot|s))}$ in (9) is still from other agents and not accessible for agent $i$. So, in the second stage, we apply (9) for all agents and rearrange the terms in $\tilde{M}\sqrt{\sum_{j\neq i}D_{\mathrm{KL}}(\pi_{\mathrm{old}}^j(\cdot|s)\|\pi^j(\cdot|s))}$ by inequalities in Theorem 3.3 to make sure that the term $D_{\mathrm{KL}}(\pi_{\mathrm{old}}^j(\cdot|s)\|\pi^j(\cdot|s))$ is optimized by agent $j$ instead of agent $i$. Finally, we have a new surrogate (12) which could be divided into $N$ parts, and each part is only related to one agent. Our general idea mentioned above is eventually realized after these two stages.

We also would like to discuss the relation between $D_{\mathrm{KL}}$ and $\sqrt{D_{\mathrm{KL}}}$ in the surrogate (12). It seems that $D_{\mathrm{KL}}$ and $\sqrt{D_{\mathrm{KL}}}$ are similar and could be merged into one term. However, in fact, we are not able to merge these two terms. From the perspective of theory, $\sqrt{D_{\mathrm{KL}}}$ is from the bound (9). The relation between $D_{\mathrm{KL}}$ and $\sqrt{D_{\mathrm{KL}}}$ is different around $D_{\mathrm{KL}} = 1$ and we cannot find an inequality to merge these two terms. From the perspective of experiments, we could find in the empirical results in Section 4.5 that eliminating either the term $\sqrt{D_{\mathrm{KL}}}$ or the term $D_{\mathrm{KL}}$ lowers the performance of DPO.

## 3.5 The Practical Algorithm

DPO is proposed with the simple idea that each agent optimizes the decentralized surrogate (17). However, we face the same trouble as TRPO in that the constant $\hat{M}$ is large and if we directly optimize this objective, then the step size of the policy update will be too small.

To settle this problem, we absorb the idea of the adaptive coefficient in PPO (Schulman et al., 2017). We use two adaptive coefficients $\beta_1^i$ and $\beta_2^i$ to replace the constant $\hat{M}$ and $C$ (Schulman et al., 2015a). Since any constant $C > 0$ is available in theory, we can also obtain an appropriate value of $C$ in an adaptive way. In practice, we will optimize the following objective:

---

**Algorithm 1** The practical algorithm of DPO

1: **for** episode $= 1$ to $M$ **do**
2:     **for** $t = 1$ to max_episode_length **do**
3:         select action $a_i \sim \pi^i(\cdot|s)$
4:         execute $a_i$ and observe reward $r$ and next state $s'$
5:         collect $\langle s, a_i, r, s' \rangle$
6:     **end for**
7:     Update decentralized critic according to (8)
8:     Update policy according to the surrogate (20)
9:     Update $\beta_1^i$ and $\beta_2^i$ according to (21).
10: **end for**

---

$$\pi_{\text{new}}^i(\cdot|s) = \arg\max_{\pi^i} \left( \frac{1}{N} \mathcal{L}_{\boldsymbol{\pi}_{\text{old}}}^i(\pi^i, s) - \beta_1^i \sqrt{D_{\text{KL}}(\pi_{\text{old}}^i(\cdot|s)\|\pi^i(\cdot|s))} - \beta_2^i D_{\text{KL}}(\pi_{\text{old}}^i(\cdot|s)\|\pi^i(\cdot|s)) \right). \quad (20)$$

As for the adaption of $\beta_1^i$ and $\beta_2^i$, we need to define a hyperparameter $d_{target}$, which can be seen as a ruler for the average KL-divergence $D_{\text{KL}}^{\text{avg}}(\pi_{\text{old}}^i\|\pi_{\text{new}}^i)$ for each agent, where $D_{\text{KL}}^{\text{avg}}(\pi_{\text{old}}^i\|\pi^i) = \mathbb{E}_{s \sim \boldsymbol{\pi}_{\text{old}}}\left[D_{\text{KL}}(\pi_{\text{old}}^i(\cdot|s)\|\pi^i(\cdot|s))\right]$.

If $D_{\text{KL}}^{\text{avg}}(\pi_{\text{old}}^i\|\pi_{\text{new}}^i)$ is close to $d_{target}$, then we believe current $\beta_1^i$ and $\beta_2^i$ are appropriate. If $D_{\text{KL}}^{\text{avg}}(\pi_{\text{old}}^i\|\pi_{\text{new}}^i)$ exceeds $d_{target}$ too much, we believe $\beta_1^i$ and $\beta_2^i$ are small and need to increase and vice versa. In practice, we will use the following rule to update $\beta_1^i$ and $\beta_2^i$:

$$\begin{aligned} &\text{If } D_{\text{KL}}^{\text{avg}}(\pi_{\text{old}}^i\|\pi_{\text{new}}^i) > d_{target} * \delta, \quad \text{then } \beta_j^i \leftarrow \beta_j^i * \omega \quad \forall j \in \{1, 2\} \\ &\text{If } D_{\text{KL}}^{\text{avg}}(\pi_{\text{old}}^i\|\pi_{\text{new}}^i) < d_{target}/\delta, \quad \text{then } \beta_j^i \leftarrow \beta_j^i/\omega \quad \forall j \in \{1, 2\}. \end{aligned} \quad (21)$$

We choose the constants $\delta = 1.5$ and $\omega = 2$ as the choice in PPO (Schulman et al., 2017). As for the critic, we just follow the standard method in PPO. Then, we can have the fully decentralized learning procedure of DPO for each agent $i$ in Algorithm 1.

The practical algorithm of DPO uses some approximations of the decentralized surrogate. Most of these approximations are traditional practices in RL or with no alternative in fully decentralized learning yet. We admit that the practical algorithm may not maintain the theoretical guarantee. However, we need to argue that we go one step further to give a decentralized surrogate in fully decentralized policy optimization with a convergence guarantee. We believe and expect that a better practical method can be found based on this objective in future work.

## 4 Experiments

In this section, we compare the practical algorithm of DPO with IPPO (de Witt et al., 2020) and IQL (Tan, 1993) in a variety of cooperative multi-agent environments, including a cooperative stochastic game, MPE (Lowe et al., 2017), multi-agent MuJoCo (Peng et al., 2021), and SMAC (Samvelyan et al., 2019), covering both discrete and continuous action spaces, as well as fully and partially observable environments. *In our experiments, we focus on the comparison between DPO and IPPO, and just add IQL as a baseline for reference. Q-learning or value-based methods are out of our scope of discussion.* As we consider fully decentralized learning, in the experiments *agents do not use parameter-sharing* as sharing parameters should be considered as centralized learning (Terry et al., 2020). In all experiments, the network architectures and common hyperparameters of DPO and IPPO are the same for a fair comparison. Note that IPPO (de Witt et al., 2020) is the widely used clip version of PPO, while the KL version of PPO serves as ablation of DPO, which is also included in the ablation study. More details about experimental settings and hyperparameters are available in Appendix A and B. Moreover, all the learning curves are from 5 random seeds and the shaded area corresponds to the 95% confidence interval.

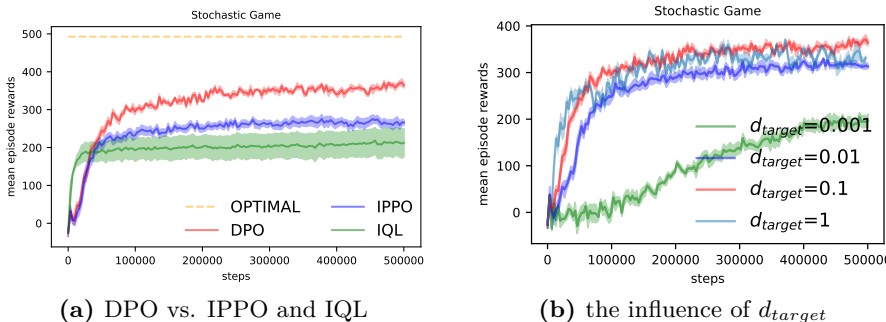

**(a)** DPO vs. IPPO and IQL

**(b)** the influence of $d_{target}$

**Figure 1:** Empirical studies of DPO on the didactic example: (a) learning curve of DPO compared with IPPO, IQL, and the global optimum; (b) the influence of different values of $d_{target}$ on DPO, x-axis is environment steps.

## 4.1 A Didactic Example

First, we use a cooperative stochastic game as a didactic example. The cooperative stochastic game has 100 states and 6 agents. Each agent has 5 actions. All the agents share a joint reward function. The reward function and the transition probability are both generated randomly. This stochastic game has a certain degree of complexity which is helpful to distinguish the performance of DPO, IPPO, and IQL. On the other hand, this environment is tabular which means training in this environment is fast and we can do ablation studies efficiently. Moreover, we can find the global optimum by dynamic programming to compare with in this game.

The learning curves in Figure 1(a) show that DPO performs better than two baselines and learns a better solution in this environment. IQL is the most unstable among the three algorithms according to its variance. The fact that DPO learns a sub-optimal solution agrees with our theoretical result. However, the sub-optimal solution found by DPO is still far from the global optimum. This means that there is still improvement space.

On the other hand, we study the influence of the hyperparameter $d_{target}$ on DPO. We choose $d_{target} = 0.001, 0.01, 0.1, 1$. The empirical results are shown in Figure 1(b). We find that when $d_{target}$ is small, the coefficients $\beta_1$ and $\beta_2$ are more likely to be increased and the step size of the policy update is limited. So for the case that $d_{target} = 0.001, 0.01$, the performance of DPO is relatively low. And when $d_{target}$ is large, the policy update may be out of the trust region. This can be witnessed by the fluctuating learning curve of the case $d_{target} = 1$. So we need to choose an appropriate value for $d_{target}$ and in this environment we choose $d_{target} = 0.1$, which is also the learning curve of DPO in Figure 1(a). We found that the appropriate value for $d_{target}$ changes in different environments. In the following, we keep $d_{target}$ to be the same for tasks of the same environment. There may be some better choices for $d_{target}$, but it is a bit time-consuming and out of the focus of our discussion.

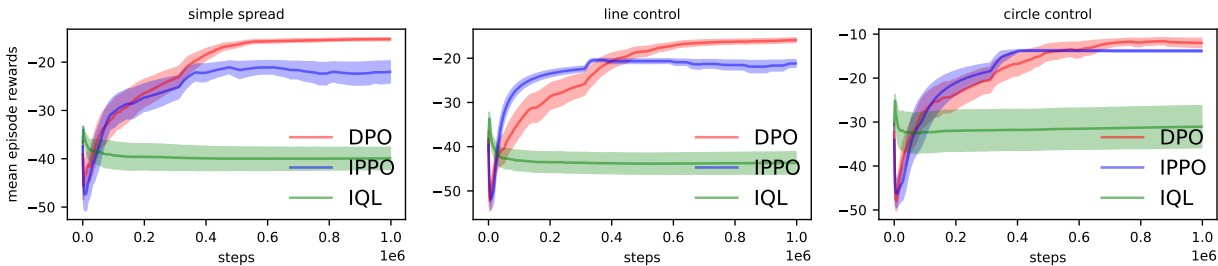

**Figure 2:** Learning curve of DPO compared with IPPO and IQL in the 5-agent simple spread, 5-agent line control, and 5-agent circle control in MPE, where the x-axis is environment steps.

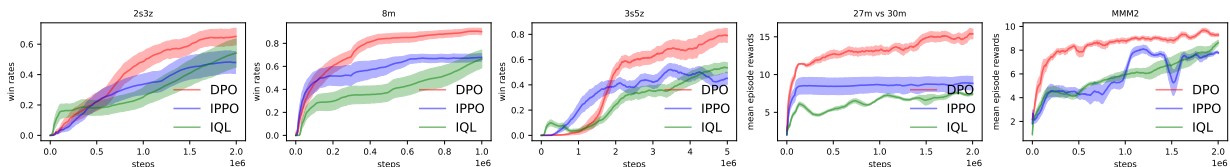

**Figure 3:** Learning curves of DPO compared with IPPO and IDDPG in $3 \times 1$ Hopper, $3 \times 2$ HalfCheetah, $3 \times 2$ Walker2d, $4 \times 2$ Ant, and $17 \times 1$ Humanoid in multi-agent MuJoCo, where x-axis is environment steps.

**Figure 4:** Learning curves of DPO compared with IPPO and IQL in 2s3z, 8m, 3s5z, 27m_vs_30m, and MMM2 in SMAC, where x-axis is environment steps.

## 4.2 MPE

MPE is a popular environment in cooperative MARL. MPE is a 2D environment and the objects in MPE environment are either agents or landmarks. Landmark is a part of the environment, while agents can move in any direction. With the relation between agents and landmarks, we can design different tasks. We use the discrete action space version of MPE and the agents can accelerate or decelerate in the direction of the x-axis or y-axis. We choose MPE for its partial observability. We take $d_{target} = 0.01$ for all MPE tasks.

The MPE tasks we used for the experiments are simple spread, line control, and circle control which were originally used in Agarwal et al. (2020). In our experiments, we set the number of agents $N = 5$ in all three tasks. The empirical results are illustrated in Figure 2. We can find that although DPO may fall behind IPPO and IQL at the beginning of the training in some tasks, DPO learns a better policy in the end for all three tasks. As for the drop in IQL's performance, we checked the learning curves of different random seeds and found that this phenomenon is actually caused by the instability of IQL. IQL learns very fast but converges to different sub-optima in different random seeds.

## 4.3 Multi-Agent MuJoCo

Multi-agent MuJoCo is a robotic locomotion control environment for multi-agent settings, which is built upon single-agent MuJoCo (Todorov et al., 2012). In multi-agent MuJoCo, each agent controls one part of a robot to carry out different tasks. We choose this environment for the reason of continuous state and action spaces. We use independent DDPG (Lillicrap et al., 2016) (IDDPG) to replace IQL for continuous action spaces. We select 5 tasks for our experiments: 3-agent Hopper, 3-agent HalfCheetah, 3-agent Walker2d, 4-agent Ant and 17-agent Humanoid. In all these tasks, we set agent_obsk=2. We take $d_{target} = 0.001$ for all multi-agent MuJoCo tasks.

The empirical results are illustrated in Figure 3. We can find that in all five tasks, DPO outperforms IPPO and IDDPG, though in 3-agent HalfCheetah DPO learns slower than IPPO at the beginning. The results on multi-agent MuJoCo verify that DPO is also effective in facing continuous state and action spaces. Moreover, the better performance of DPO in the 17-agent Humanoid task could be evidence of the scalability of DPO.

## 4.4 SMAC

SMAC is a partially observable and high-dimensional environment that has been used in many cooperative MARL studies. We select five maps in SMAC, 2s3z, 8m, 3s5z, 27m_vs_30m, and MMM2 for our experiments. We take $d_{target} = 0.02$ for all SMAC tasks.

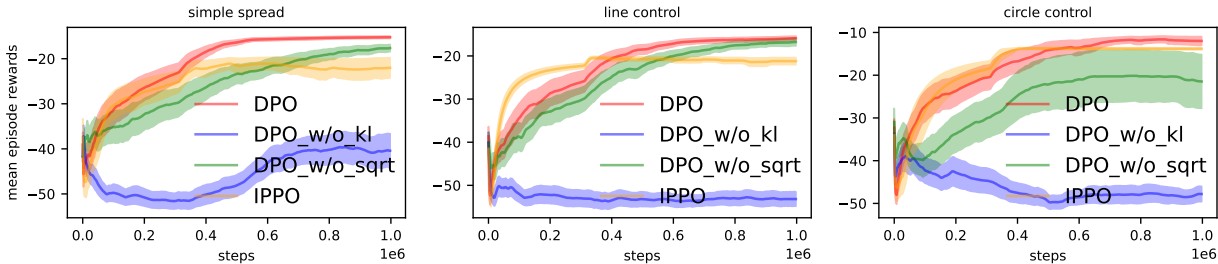

**Figure 5:** Learning curves of DPO compared with DPO_w/o_kl and DPO_w/o_sqrt in three MPE tasks, where x-axis is environment steps.

The empirical results are illustrated in Figure 4. The two super hard SMAC tasks (27m_vs_30m and MMM2) are too difficult for all DPO, IPPO, and IQL to win, so we use episode reward as the metric to show their difference. DPO performs better than IPPO and IQL in all five maps. We need to argue that though we have controlled the network architectures of DPO and IPPO to be the same, in our experiments each agent has its individual parameters which increases the difficulty of training. So our results in SMAC may be different from other works. Although IPPO has been shown to perform well in SMAC (de Witt et al., 2020; Yu et al., 2021; Papoudakis et al., 2021), DPO can still outperform IPPO, which verifies the effectiveness of the practical algorithm of DPO in high-dimensional complex tasks and can also be evidence of our theoretical result. Again, the better performance of DPO in 27m_vs_30m shows its good scalability in the task with many agents. As for the performance of IPPO, we have fine-tuned the clip parameters and found that the impact of different clip parameters is relatively small. The empirical results and more discussions are included in Appendix C.

### 4.5 Ablation Study

We carry out the ablation study about the objective in (20). We consider two ablation methods: in the first one, we keep $\beta_1^i = 0$ to eliminate the influence of the term $\sqrt{D_{\mathrm{KL}}^{\mathrm{avg}}(\pi_{\mathrm{old}}^i\|\pi^i)}$, which is the same as IPPO-KL; in the second one, we keep $\beta_2^i = 0$ to eliminate the influence of the term $D_{\mathrm{KL}}^{\mathrm{avg}}(\pi_{\mathrm{old}}^i\|\pi^i)$. The other parameters are controlled to be the same as DPO. We call these two methods as DPO_w/o_sqrt and DPO_w/o_kl, respectively.

We compare DPO with DPO_w/o_kl and DPO_w/o_sqrt for ablation study in MPE, multi-agent MuJoCo, and SMAC. We add the performance of IPPO in the empirical results for reference. The tasks selected for each environment are the same as previous experiments in Section 4.2, 4.3 and 4.4. The empirical results for MPE, multi-agent MuJoCo, and SMAC are illustrated in Figure 5, Figure 6, and Figure 7, respectively. In general, we can find that eliminating either the term $\sqrt{D_{\mathrm{KL}}^{\mathrm{avg}}(\pi_{\mathrm{old}}^i\|\pi^i)}$ or the term $D_{\mathrm{KL}}^{\mathrm{avg}}(\pi_{\mathrm{old}}^i\|\pi^i)$ lowers the performance of DPO in all these tasks. Though the influence of eliminating these two terms is relatively small in several tasks such as Walker2d-V2_3X2 in multi-agent MuJoCo and 2s3z in SMAC, the absence of these two terms will obviously lower the performance of DPO in most tasks. This could be evidence for the significance of our novel lower bound (17) and the objective (20). The performance drop also shows empirically that we are not able to merge the two KL-divergence terms into one.

## 5 Conclusion

In this paper, we investigate fully decentralized learning in cooperative multi-agent reinforcement learning. We derive a novel decentralized lower bound for the joint policy improvement and we propose DPO, a fully decentralized actor-critic algorithm with convergence guarantee and monotonic improvement. Empirically, we test DPO compared with IPPO and IQL in a variety of environments including a cooperative stochastic game, MPE, multi-agent MuJoCo, and SMAC, covering both discrete and continuous action spaces, as well as fully and partially observable environments. The results show the advantage of DPO over IPPO and IQL, which can be evidence for our theoretical results.

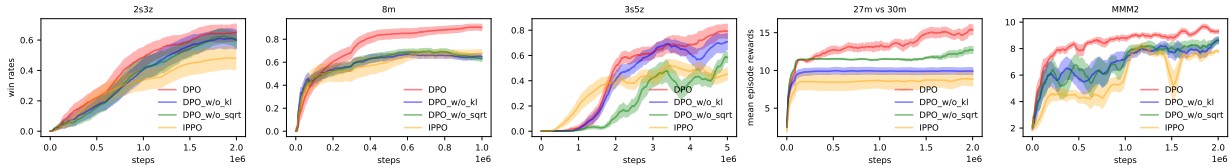

**Figure 6:** Learning curves of DPO with compared with DPO_w/o_kl and DPO_w/o_sqrt in 3-agent Hopper, 3-agent HalfCheetah, 3-agent Walker2d, 4-agent Ant, and 17-agent Humanoid in multi-agent MuJoCo, where x-axis is environment steps.

**Figure 7:** Learning curves of DPO with compared with DPO_w/o_kl and DPO_w/o_sqrt on 2s3z, 8m,3s5z, MMM2, and 27m_vs_30m in SMAC , where x-axis is environment steps.

We have to admit that there are still some limitations to DPO. To optimize the decentralized surrogate objective, the practical algorithm of DPO needs to take some approximations, which however may not preserve the theoretical results. In future work, we hope to find a better practical method for our theoretical results.

**Acknowledgements**

This work was supported by NSF China under grant 62250068. The authors would like to thank the anonymous reviewers for their valuable comments.

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

# A   Experimental Settings

## A.1   MPE

The three tasks are built on the origin MPE (Lowe et al., 2017) (MIT license) and are originally used in Agarwal et al. (2020) (MIT license). The objectives in these three tasks are listed as follows:

- **Simple Spread:** There are $N$ agents who need to occupy the locations of $N$ landmarks.
- **Line Control:** There are $N$ agents who need to line up between 2 landmarks.
- **Circle Control:** There are $N$ agents who need to form a circle around a landmark.

The reward in these tasks is the distance between all the agents and their target locations. We set the number of agents $N = 5$ for these three tasks in our experiment.

## A.2   Multi-Agent MuJoCo

Multi-agent MuJoCo (Peng et al., 2021) (Apache-2.0 license) is a robotic locomotion task with continuous action space for multi-agent settings. MuJoCo's reward function is about the distance the robot has moved from the original position. In Multi-Agent MuJoCo, the robot could be divided into several parts and each part contains several joints. Agents in this environment control a part of the robot which could be different varieties. So the type of the robot and the assignment of the joints decide a task. For example, the task 'HalfCheetah-3×2' means dividing the robot 'HalfCheetah' into three parts for three agents and each part contains 2 joints.

The details about our experiment settings in multi-agent Mujoco are listed in Table 1. The configuration defines the number of agents and the joints of each agent. The 'agent obsk' defines the number of nearest agents an agent can observe.

**Table 1:** The task settings of multi-agent MuJoCo

| task | configuration | agent obsk |
|------|:---:|:---:|
| HalfCheetah | 3×2 | 2 |
| Hopper | 3×1 | 2 |
| Walker2d | 3×2 | 2 |
| Ant | 4×2 | 2 |
| Humanoid | 17×1 | 2 |

## A.3   SMAC

SMAC (Samvelyan et al., 2019) is a popular environment for MARL. The agents in SMAC are rewarded as soon as they attack or kill an enemy unit. The rewards for an episode in SMAC are affected by the number of agents, so the environment has normalized the maximum episode rewards for all tasks to 20. If the agents in SMAC kill all enemy units in an episode, then they have 'won" that episode. The observation space of the agents in SMAC is related to the number of units in the task. In general, the observation is a vector with 100+ dimensions over the information of all units in difficult tasks, and the information of the units that are outside the agent's field of view is denoted by zero in the observation vector. More details on SMAC can be found in the original paper (Samvelyan et al., 2019).

# B   Training Details

Our code of IPPO is based on the open-source code[1] of MAPPO (Yu et al., 2021) (MIT license). We modify the code for individual parameters and ban the tricks used by MAPPO for SMAC. The network architectures

---

[1]https://github.com/marlbenchmark/on-policy

and base hyperparameters of DPO and IPPO are the same for all the tasks in all the environments. We use 3-layer MLPs for the actor and the critic and use ReLU as non-linearities. The number of the hidden units of the MLP is 128. We train all the networks with an Adam optimizer. The learning rates of the actor and critic are both 5e-4. The number of epochs for every batch of samples is 15 which is the recommended value in Yu et al. (2021). For IPPO, the clip parameter is 0.2 which is the same as Schulman et al. (2017). For DPO, the initial values of the coefficient $\beta_1^i$ and $\beta_2^i$ are 0.01. The value of $d_{\text{target}}$ is 0.1 for the cooperative stochastic game, 0.01 for MPE, 0.001 for multi-agent MuJoCo, and 0.02 for SMAC. Our code of IQL is based on the open-source code[2] PyMARL (Apache-2.0 license) and we modify the code for individual parameters. The default architecture in PyMARL is RNN so we just follow it and the number of the hidden units is 128. The learning rate of IQL is also 5e-4. The architectures of the actor and critic of IDDPG are 3-layer MLPs. The learning rates of the actor and critic are both 5e-4.

**Table 2:** Hyperparameters for all the experiments

| hyperparameter | value |
| --- | --- |
| MLP layers | 3 |
| hidden size | 128 |
| non-linear | ReLU |
| optimizer | Adam |
| actor_lr | 5e-4 |
| critic_lr | 5e-4 |
| numbers of epochs | 15 |
| initial $\beta_1^i$ | 0.01 |
| initial $\beta_2^i$ | 0.01 |
| $\delta$ | 1.5 |
| $\omega$ | 2 |
| $d_{\text{target}}$ | different for environments as aforementioned |
| clip parameter for IPPO | 0.2 |

The version of the game StarCraft2 in SMAC is 4.10 for our experiments in all the SMAC tasks. We set the episode length of all the multi-agent MuJoCo tasks as 1000 in all of our multi-agent MuJoCo experiments. We performed the whole experiment with a total of four NVIDIA A100 GPUs. We have summarized the hyperparameters in Table 2.

## C  Additional Results

Schulman et al. (2017) actually proposed two versions of PPO. The first version, which is also the most popular version, is with the clip trick. The second version is directly optimizing the penalty formula with adaptive coefficients and we refer to this algorithm as PPO-KL. IPPO (de Witt et al., 2020) is actually extended from the first version, while the practical algorithm of DPO is similar to the second version. The main difference between DPO and PPO-KL is the term of the square root of the KL-divergence in the policy loss. We modify IPPO by making each agent learn with PPO-KL to obtain IPPO-KL.

For the completeness of our experiments, we test the performance of IPPO-KL, IQL, and DPO_w/o_kl that eliminates the term $D_{\text{KL}}^{\text{avg}}(\pi_{\text{old}}^i \| \pi^i)$ for ablation study, in the cooperative stochastic game. The empirical result is illustrated in Figure 8. We find that the performances of IPPO-KL, IPPO, and DPO_w/o_kl are close and are all lower than DPO. This could be evidence of the effectiveness of DPO. As for IQL, its performance is lower than IPPO though it converges faster.

Moreover, we further study the influence of the values of $\delta$ and $\omega$ for the adaptive adjustments of the coefficient $\beta_1^i$ and $\beta_2^i$. We test seven different choices of $\delta$ and $\omega$ as $(\delta, \omega) = (1.5, 2), (1.5, 4), (1.5, 6), (1.1, 2), (3, 2), (3, 6), (1.1, 6)$ in the cooperative stochastic game. The empirical results are included in Figure 9. We find that the influence of different $\delta$ and $\omega$ is relatively limited and the adaptive adjustment is not very sensitive to them. This conclusion is similar to the PPO paper (Schulman et al., 2017).

---

[2]https://github.com/oxwhirl/pymarl

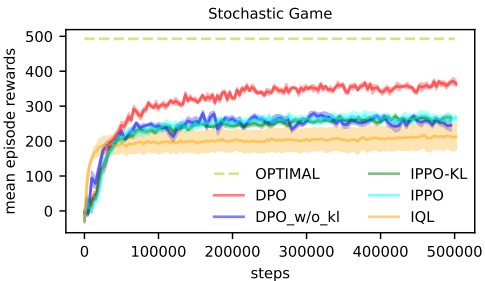

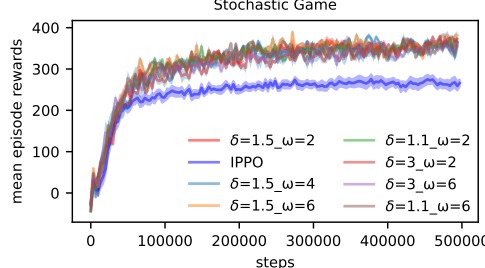

**Figure 8:** Learning curve of DPO compared with DPO_w/o_kl, IPPO, IPPO-KL, IQL, and the global optimum in the cooperative stochastic game, where the x-axis is environment steps.

**Figure 9:** Learning curves of DPO with different values of $\delta$ and $\omega$ in the cooperative stochastic game, where the x-axis is environment steps.

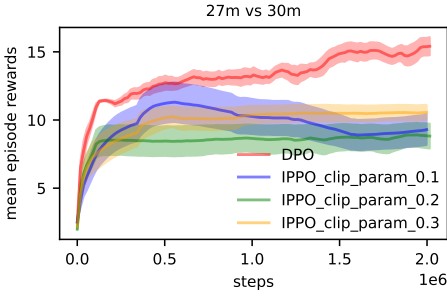

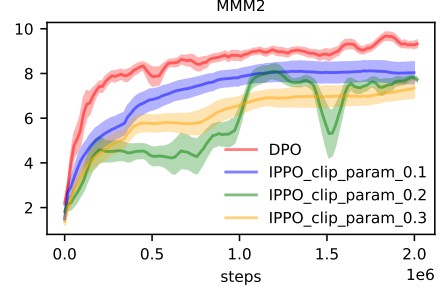

**Figure 10:** Learning curves of DPO with compared with IPPO with different clip parameters in 27m_vs_30m and MMM2 in SMAC, where x-axis is environment steps.

IPPO performs poorly in super-hard SMAC tasks such as 27m_vs_30m and MMM2 where the number of agents is relatively more and IPPO can hardly win. So we finetune the clip parameter of IPPO in these tasks to find the impact of hyperparameters on IPPO. The empirical results are illustrated in Figure 10. We could find that different clip parameters only affect the performance of IPPO during the training process, but at the end of the training, these algorithms perform similarly. We need to argue again here that in our settings parameter sharing is not permitted so all the methods will have difficulties in training as the number of agents increases.

To determine whether the bound in (9) is trivial in theory is actually not simple. But we need to argue that the bound will be tighter if the KL-divergence between the new policies and the old policies is smaller. So we empirically evaluate the mean KL-divergence of all agents in the training process in two SMAC tasks, 27m_vs_30m and MMM2. The empirical results are illustrated in Figure 12. We could find that the KL-divergence is actually small and stable during the whole learning process, which could be evidence for the effectiveness of our bound.

**Table 3:** The two-player matrix game where each agent has two actions. $p$ and $q$ represent the policies of two agents respectively. $a$, $b$, $c$ and $d$ are the rewards for the corresponding joint actions respectively.

|         | $q$ | $1-q$ |
|---------|-----|-------|
| $p$     | $a$ | $b$   |
| $1-p$   | $c$ | $d$   |

To verify the monotonic improvement property of (17), we use a simple two-player matrix game in Table 3. In this matrix game, each agent has two actions and the reward for four joint actions are $a$, $b$, $c$ and $d$ respectively. To obtain a more general result, we take four different sets of the payoff matrix: Matrix Game 1 $(a, b, c, d) = (5, 7, 6, 4)$; Matrix Game 2 $(a, b, c, d) = (1, 3, 5, 4)$; Matrix Game 3 $(a, b, c, d) = (7, 1, 1, 3)$; Matrix Game 4 $(a, b, c, d) = (20, 0, 0, 10)$. Since there is no closed-form solution to (17), we use a numerical method to find the solution for policy updates. The constant $\hat{M}$ is defined by the payoff matrix and the constant $C$ can be chosen arbitrarily. We choose $C = 10$ in all the experiments. The empirical results are illustrated in Figure 11. We use the payoff matrix and the joint policy $\boldsymbol{\pi}_t = (p_t, q_t)$ to calculate the value of $J(\boldsymbol{\pi}_t)$. We can find the phenomenon of monotonic improvement in all four matrix games. Moreover, we can find that the joint policy $\boldsymbol{\pi}_t = (p_t, q_t)$ converges in all four matrix games. From the results in Matrix Game 1 and Matrix

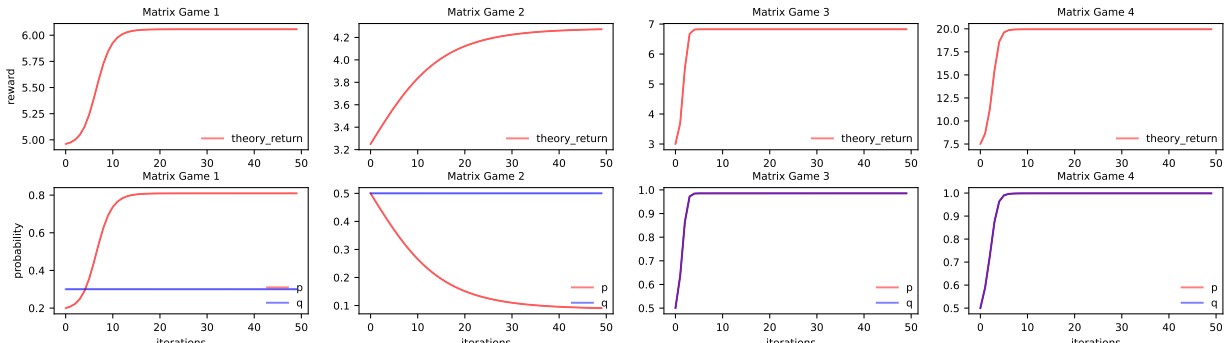

**Figure 11:** The learning curves of the iteration (17) in four different matrix game. Each column corresponds to one matrix game. The first line is the learning curves of $J(\boldsymbol{\pi}_t)$. The second line is the learning curves of the policies $p$ and $q$.

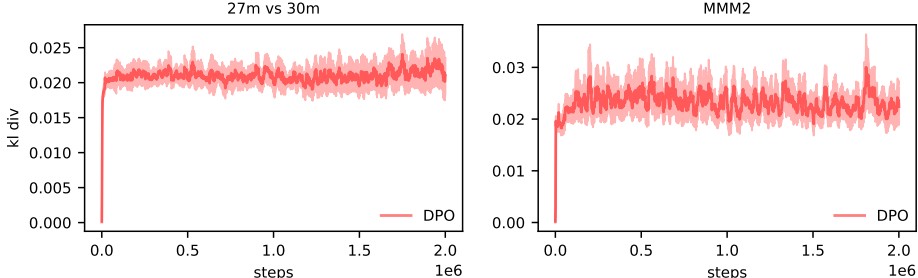

**Figure 12:** The mean KL-divergence curves of all agents for DPO in 27m__vs__30m and MMM2 in SMAC, where the x-axis is environment steps.

Game 2 we know that the iteration (17) may be trapped in the sub-optimum. These results can empirically verify our theoretical results in Theorem 3.4.

## D    Discussion

Besides our empirical results, we would like to share our views on the difference between DPO and IPPO and give some intuitive ideas. KL regularization and ratio clipping are similar in the single-agent setting, but they are not supposed to be similar in multi-agent settings. The 'correct' ratio clipping in the multi-agent setting according to the theory of PPO should clip over the joint policy ratio $\frac{\boldsymbol{\pi}_{\mathbf{new}}(\boldsymbol{a}|s)}{\boldsymbol{\pi}_{\mathbf{old}}(\boldsymbol{a}|s)}$. IPPO just clips individual policy ratio $\frac{\pi^i_{\mathrm{new}}(a_i|s)}{\pi^i_{\mathrm{old}}(a_i|s)}$ for each agent $i$ which may not be enough to realize the 'correct' ratio clipping. We could find more discussion about this in the CoPPO (Wu et al., 2021) paper. So IPPO is not supposed to enjoy the theoretical results of DPO.

We could rewrite the objective of IPPO for each agent $i$ with a similar formulation in HPO (Yao et al., 2021) as follows:

$$\mathcal{L}^{i,\mathrm{IPPO}}_{\boldsymbol{\pi}_{\mathrm{old}}}(\pi^i) = \sum_s \boldsymbol{\rho}_{\mathrm{old}}(s) \sum_{a_i} \pi^i(a_i|s)|A^i_{\mathrm{old}}(s,a_i)| l\left(\mathrm{sign}(A^i_{\mathrm{old}}(s,a_i)), u_i(s,a_i) - 1, \epsilon\right),$$

where $l(y,x,\epsilon) = \max\{0, \epsilon - y \times x\}$ is the hinge loss and $u_i(s,a_i) = \frac{\pi^i(a_i|s)}{\pi^i_{\mathrm{old}}(a_i|s)}$ is the ratio.

If we follow the same idea as PPO, then IPPO is the 'correct' ratio clipping version for the surrogate of DPO. However, the effectiveness of this ratio clipping formulation in theory is still open in decentralized learning since there is not any convergence guarantee for IPPO, to the best of our knowledge.

Though the effectiveness of IPPO, in theory, is beyond the scope of our paper, we could provide an intuitive explanation for the fact that the performance of DPO can surpass IPPO from this formulation and the

analysis in HPO. In the proof of HPO, there is a critical assumption that the sign of the estimated advantage is the same as that of the true advantage (Assumption 4 in Section 2.3 in Yao et al. (2021)). And HPO also shows that the sign of the advantage is more important than the value for this formulation of PPO-clip. In decentralized learning, both DPO and IPPO are facing the difficulty of learning the individual advantage function as there may be noise in the individual value function. However, the objective of DPO is continuous and the objective of IPPO is discrete for $\text{sign}(A^i_{\text{old}}(s, a_i))$. So the impact of the noise in the value function may be larger on IPPO than DPO.

Recently, there have been several works with theoretical analysis of monotonic improvement in MARL including HAPPO (Kuba et al., 2021), CoPPO (Wu et al., 2021) and A2PO (Wang et al., 2023b). Unfortunately, these algorithms are not appropriate for fully decentralized learning. HAPPO needs agents to maintain a joint advantage function over the joint action and the agents need the information of the critical function $M^{i_{1:m}}$ from other agents. These conditions cannot be satisfied in fully decentralized learning. CoPPO needs the ratios of other agents' policies for policy updates but this information cannot be obtained in fully decentralized learning. A2PO uses the joint advantage function and agents need the joint policy in the off-policy correction for the calculation of $A^{\boldsymbol{\pi}, \hat{\boldsymbol{\pi}}^{i-1}}$. Moreover, A2PO uses the double-clip trick similar to CoPPO which requires the ratios of other agents' policies. Thus, A2PO is also not appropriate for fully decentralized learning.

## E    Additional Proof for Decentralized Critic

In this section, we will discuss the convergence of the iteration of decentralized critic following the idea of Melo (2001). We have a lemma from Melo (2001).

**Lemma E.1.** *The random process* $\{\Delta_t\}$ *taking values in* $\mathbb{R}^n$ *and defined as*

$$\Delta_{t+1}(x) = (1 - \alpha_t(x)) + \alpha_t(x)F_t(x)$$

*converges to zero w.p.1 under the following assumptions:*

- $0 \le \alpha_t \le 1$, $\sum_t \alpha_t(x) = \infty$ *and* $\sum_t \alpha_t^2(x) < \infty$;
- $\|\mathbb{E}\left[F_t(x)|\mathcal{F}_t\right]\| \le \gamma\|\Delta_t\|_W$, *with* $\gamma < 1$;
- *$\boldsymbol{Var}\left[F_t(x)|\mathcal{F}_t\right] \le C\left(1 + \|\Delta_t\|_W^2\right)$, for $C > 0$.*

Next, we define an update rule given any initial Q-function $Q_0^i$ for any agent $i$ and the joint policy $\boldsymbol{\pi}$ as follows:

$$Q_{t+1}^i(s_t, a_t^i) = Q_t^i(s_t, a_t^i) + \alpha_t^i(s_t, a_t^i)\left(r_i(s_t, a_t^i) + \gamma\mathbb{E}_{b_t^i \sim \pi_i(\cdot|s_{t+1})}\left[Q_t^i(s_{t+1}, b_t^i)\right] - Q_t^i(s_t, a_t^i)\right), \qquad (22)$$

where $0 \le \alpha_t^i(s_t, a_t^i) \le 1$ is the step-size for agent $i$ and the reward $r_i(s, a_i) = r_{\pi^{-i}}(s, a_i) = \mathbb{E}_{\pi^{-i}}[r(s, a_i, a_{-i})]$.

Then we have the following result.

**Proposition E.2.** *Given any initial Q-function $Q_0^i$ for any agent $i$, the update rule (22) converges to $Q_{\pi^{-i}}^{\pi^i}(s, a_i)$ w.p.1 as long as*

$$\sum_t \alpha_t^i(s, a_i) = \infty \text{ and } \sum_t \alpha_t^{i2}(s, a_i) < \infty \qquad (23)$$

*for all* $(s, a_i) \in S \times A_i$.

*Proof.* We can rewrite (22) as follows

$$Q_{t+1}^i(s_t, a_t^i) = \left(1 - \alpha_t^i(s_t, a_t^i)\right)Q_t^i(s_t, a_t^i) + \alpha_t^i(s_t, a_t^i)\left(r_i(s_t, a_t^i) + \gamma\mathbb{E}_{b_t^i \sim \pi_i(\cdot|s_{t+1})}\left[Q_t^i(s_{t+1}, b_t^i)\right]\right). \qquad (24)$$

We define $\Delta_t^i(s, a_i) = Q_t^i(s, a_i) - Q_{\pi^{-i}}^{\pi^i}(s, a_i)$, then we have

$$\Delta_{t+1}^i(s_t, a_t^i) = \left(1 - \alpha_t^i(s_t, a_t^i)\right)\Delta_t^i(s_t, a_t^i) + \alpha_t^i(s_t, a_t^i)\left(r_i(s_t, a_t^i) + \gamma\mathbb{E}_{b_t^i \sim \pi_i(\cdot|s_{t+1})}\left[Q_t^i(s_{t+1}, b_t^i)\right] - Q_{\pi^{-i}}^{\pi^i}(s_t, a_t^i)\right) \qquad (25)$$

If we define $F_t^i(s, a_i) = r_i(s, a_i) + \gamma \mathbb{E}_{b_i \sim \pi_i(\cdot|s')} \left[ Q_t^i(s', b_i) \right] - Q_{\pi^{-i}}^{\pi^{-i}}(s, a_i)$, then we have

$$\Delta_{t+1}^i(s_t, a_t^i) = \left(1 - \alpha_t^i(s_t, a_t^i)\right) \Delta_t^i(s_t, a_t^i) + \alpha_t^i(s_t, a_t^i) F_t^i(s_t, a_t^i) \tag{26}$$

To apply Lemma E.1, we consider $\mathbb{E}\left[F_t^i(s, a_i)|\mathcal{F}_t\right]$ and $\mathbf{Var}\left[F_t^i(s, a_i)|\mathcal{F}_t\right]$.

For $\mathbb{E}\left[F_t^i(s, a_i)|\mathcal{F}_t\right]$, we have

$$
\begin{aligned}
\mathbb{E}\left[F_t^i(s, a_i)|\mathcal{F}_t\right] &= \mathbb{E}_{s' \sim P_i(\cdot|s, a_i)} \left[ r_i(s, a_i) + \gamma \mathbb{E}_{b_i \sim \pi_i(\cdot|s')} \left[ Q_t^i(s', b_i) \right] - Q_{\pi^{-i}}^{\pi^{-i}}(s, a_i) \right] \\
&= r_i(s, a_i) + \gamma \mathbb{E}_{s' \sim P_i(\cdot|s, a_i), b_i \sim \pi_i(\cdot|s')} \left[ Q_t^i(s', b_i) \right] - Q_{\pi^{-i}}^{\pi^{-i}}(s, a_i) \\
&= \Gamma_{\pi^{-i}}^{\pi^i} Q_t^i(s, a_i) - Q_{\pi^{-i}}^{\pi^{-i}}(s, a_i) \tag{27} \\
&= \Gamma_{\pi^{-i}}^{\pi^i} Q_t^i(s, a_i) - \Gamma_{\pi^{-i}}^{\pi^i} Q_{\pi^{-i}}^{\pi^{-i}}(s, a_i), \tag{28}
\end{aligned}
$$

where $P_i(s'|s, a_i) = \sum_{a_{-i}} \pi_{-i}(a_{-i}|s) P(s'|s, a_i, a_{-i})$ is the individual transition probability from the perspective of agent $i$ given a fixed $\pi_{-i}$. The step (27) is from the definition of $\Gamma_{\pi^{-i}}^{\pi^i}$ in (5). The step (28) is from the property of the fixed point in (4). From the contraction property of $\Gamma_{\pi^{-i}}^{\pi^i}$ in (6), we know

$$\left\| \mathbb{E}\left[F_t^i(s, a_i)|\mathcal{F}_t\right] \right\|_\infty = \left\| \Gamma_{\pi^{-i}}^{\pi^i} Q_t^i - \Gamma_{\pi^{-i}}^{\pi^i} Q_{\pi^{-i}}^{\pi^{-i}} \right\|_\infty \leq \gamma \left\| Q_t^i - Q_{\pi^{-i}}^{\pi^{-i}} \right\|_\infty = \gamma \left\| \Delta_t^i \right\|_\infty \tag{29}$$

For $\mathbf{Var}\left[F_t^i(s, a_i)|\mathcal{F}_t\right]$, we have

$$
\begin{aligned}
\mathbf{Var}\left[F_t^i(s, a_i)|\mathcal{F}_t\right] &= \mathbb{E}\left[ r_i(s, a_i) + \gamma \mathbb{E}_{b_i \sim \pi_i(\cdot|s')} \left[ Q_t^i(s', b_i) \right] - Q_{\pi^{-i}}^{\pi^{-i}}(s, a_i) - \left( \Gamma_{\pi^{-i}}^{\pi^i} Q_t^i(s, a_i) - Q_{\pi^{-i}}^{\pi^{-i}}(s, a_i) \right) \right] \\
&= \mathbb{E}\left[ r_i(s, a_i) + \gamma \mathbb{E}_{b_i \sim \pi_i(\cdot|s')} \left[ Q_t^i(s', b_i) \right] - \Gamma_{\pi^{-i}}^{\pi^i} Q_t^i(s, a_i) \right] \\
&= \mathbf{Var}\left[ r_i(s, a_i) + \gamma \mathbb{E}_{b_i \sim \pi_i(\cdot|s')} \left[ Q_t^i(s', b_i) \right] |\mathcal{F}_t \right].
\end{aligned}
$$

Given the fact that $r_i$ is bounded, we know that $\mathbf{Var}\left[F_t^i(s, a_i)|\mathcal{F}_t\right] \leq C\left(1 + \left\| \Delta_t^i \right\|_\infty\right)$ for some constant $C$. Finally from the Lemma E.1, we know $\{\Delta_t^i\}$ converges to zero w.p.1, *i.e.*, $\{Q_t^i\}$ converges to $Q_{\pi^{-i}}^{\pi^{-i}}$ w.p.1. $\quad\square$

# F   Future Work

In the paper, we derive a novel lower bound that can be naturally divided into independent surrogate (17) for each agent. By each agent optimizing this surrogate, the monotonic improvement of the joint policy can be guaranteed in fully decentralized settings. However, the practical algorithm of DPO takes the formula of (20) with several approximations. How to solve the optimization of (17) more precisely is left as future work. Moreover, we expect our work could provide some insights for future studies on fully decentralized multi-agent reinforcement learning since current methods still have a gap from the optimum as shown in Figure 8.

