# OpenReview forum: "A Fully Decentralized Surrogate for Multi-Agent Policy Optimization"
_TMLR — Accepted by TMLR_

### Review · Reviewer_ToSz · 2023-08-29

**Summary Of Contributions:**

This paper provides a decentralized policy optimization algorithm (DPO) with convergence and monotonic improvement guarantees. Extensive experiments in different types of cooperative MARL testbeds show good performances of DPO.

**Audience:**

No

**Claims And Evidence:**

No

**Requested Changes:**

Please address my points on weaknesses above.

**Strengths And Weaknesses:**

Overall this is a good paper that provides a detailed theoretical study of decentralized policy optimization. Further, it contains a practical algorithm with detailed experimental results in different multi-agent settings. Despite this, I have some important comments regarding the weaknesses of the paper.

1) In the introduction the motivation says that "many settings in which global information is unavailable", which led me to believe that the authors are studying partially observable settings. However all the theory in the paper uses the global state and not the observation of individual agents. The experiments seems to use partially observable settings (from the configurations in the appendix). So, I am confused what the authors are trying to accomplish and what setting they are using the paper.

2) There are some important details missing in the paper to ensure reproducibility. First, the reward functions of some environments (Mujoco and SMAC) are not mentioned anywhere in the paper (main paper or appendix), so I am not able to fully interpret the results with the mean episodic rewards. Second, while SMAC is mentioned to be partially observable, how big is the state space and the observation space of agents in this environment, as used in the experiments of the paper? Also, what conditions are needed for "winning" as plotted in figure 7 of the paper.

3) Several methods in the experiments show unlearning (ex: IDDPG in Ant, DPO w/o Sqrt in circle control etc.). This makes me feel that the DPO algorithm could show unlearning if run for more steps in the future. Do the authors have any intuitions for why this may not be the case? If not, I would recommend experiments to be conducted for longer periods. Especially several methods in the paper in all experiments still seem to be improving/changing and have not reached a stable point in training.

4) In the experiments different performances of algorithms in different experiments do not have clear explanations. For instance, why does IQL show instability in MPE, but performance close to the best performing algorithm in SMAC? This may show that the hyper-parameters have not been carefully tuned for these baselines in all experiments.

5) In general, I am confused as to what the experiments are trying to accomplish. The paper says that "focus on comparison between DPO and IPPO and just add IQL as baseline". Why are such comparisons helpful? Also, I strongly disagree with the conclusion in the paper "the results show the advantage of DPO over IPPO and IQL, which can be evidence for our theoretical results" -- the algorithm has important differences with respect to the theory and I am not convinced it is acting as an evidence for any theoretical findings.

6) There seems to be some abuse of notations in the theory which makes it hard to follow everything. The objective $J(\pi)$ is not conditioned on the initial state (though it should be conditioned on some initial state). The policy (and joint policy) was never defined. In some cases the Q function and value function are not parameterized by the agent index (Eq for $A^i_{old}$ following Eq.6), and in other cases they are parameterized by the agent index (Eq 7), which makes me more confused.

---

> ### Author Response · Authors · 2023-11-17
> **Response**
>
> > In the introduction the motivation says that "many settings in which global information is unavailable", which led me to believe that the authors are studying partially observable settings. However all the theory in the paper uses the global state and not the observation of individual agents. The experiments seems to use partially observable settings (from the configurations in the appendix). So, I am confused what the authors are trying to accomplish and what setting they are using the paper.
>
> We use the global state $s$ only for theoretical analysis, just as previous work containing theoretical results in decentralized learning, which include both cooperative settings [1] and non-cooperative settings [2, 3, 4]. The difficulty of solving a POMDP has been studied for decades in [5, 6, 7], and the theoretical analysis of Dec-POMDP will be even more difficult in the multi-agent setting. If we include partial observability in the analysis, we may obtain nothing since the problem may be undecidable in Dec-POMDP.
>
> On the other hand, we have to admit that partial observability is important for some applications in MARL and many popular multi-agent environments are partially observable. In the experiments, we will allow agents to use their individual observations or history to approximate the state. Empirically, DPO still works in the partially observable tasks. Our meaning of the global information includes the actions of other agents.
>
> [1] Jiechuan Jiang and Zongqing Lu. I2q: A fully decentralized q-learning algorithm. In Advances in Neural
> Information Processing Systems (NeurIPS), 2022.
>
> [2] Gürdal Arslan and Serdar Yüksel. Decentralized q-learning for stochastic teams and games. IEEE Transac-
> tions on Automatic Control, 62(4):1545–1558, 2016.
>
> [3] Weichao Mao, Lin Yang, Kaiqing Zhang, and Tamer Basar. On improving model-free algorithms for de-
> centralized multi-agent reinforcement learning. In International Conference on Machine Learning, pp.
> 15007–15049. PMLR, 2022.
>
> [4] Runyu Zhang, Zhaolin Ren, and Na Li. Gradient play in stochastic games: stationary points, convergence,
> and sample complexity. arXiv preprint arXiv:2106.00198, 2021b
>
> [5] Christos H Papadimitriou and John N Tsitsiklis. The complexity of markov decision processes. Mathematics
> of operations research, 12(3):441–450, 1987.
>
> [6] Martin Mundhenk, Judy Goldsmith, Christopher Lusena, and Eric Allender. Complexity of finite-horizon
> markov decision process problems. Journal of the ACM (JACM), 47(4):681–720, 2000.
>
> [7] Nikos Vlassis, Michael L Littman, and David Barber. On the computational complexity of stochastic con-
> troller optimization in pomdps. ACM Transactions on Computation Theory (TOCT), 4(4):1–8, 2012.
>
> > There are some important details missing in the paper to ensure reproducibility. First, the reward functions of some environments (Mujoco and SMAC) are not mentioned anywhere in the paper (main paper or appendix), so I am not able to fully interpret the results with the mean episodic rewards. Second, while SMAC is mentioned to be partially observable, how big is the state space and the observation space of agents in this environment, as used in the experiments of the paper? Also, what conditions are needed for "winning" as plotted in figure 7 of the paper.
>
> Thank you for pointing out this missing information. MuJoCo's reward function is about the distance the robot has moved from the original position. The agents in SMAC are rewarded as soon as they attack or kill an enemy unit. The rewards for an episode in SMAC are affected by the number of agents, so the environment has normalized the maximum episode rewards for all tasks to 20. If the agents in SMAC kill all enemy units in an episode, then they have 'won" that episode. The observation space of the agents in SMAC is related to the number of units in the task. In general, the observation is a vector with 100+ dimensions over the information of all units in difficult tasks, and the information of the units that are outside the agent's field of view is denoted by zero in the observation vector. More details on these two environments can be found in the original paper [1, 2]. We believe that these environments are very popular in the MARL community, so we omit this introduction in this paper. We have added more explanations of these environments in Appendix A.
>
>
>
> [1] Mikayel Samvelyan, Tabish Rashid, Christian Schroeder de Witt, Gregory Farquhar, Nantas Nardelli, Tim GJ
> Rudner, Chia-Man Hung, Philip HS Torr, Jakob Foerster, and Shimon Whiteson. The starcraft multi-agent
> challenge. arXiv preprint arXiv:1902.04043, 2019.
>
> [2] Bei Peng, Tabish Rashid, Christian Schroeder de Witt, Pierre-Alexandre Kamienny, Philip Torr, Wendelin
> Böhmer, and Shimon Whiteson. Facmac: Factored multi-agent centralised policy gradients. In Advances in
> Neural Information Processing Systems (NeurIPS), 2021.

---

> ### Author Response · Authors · 2023-11-17
> **Response**
>
> > Several methods in the experiments show unlearning (ex: IDDPG in Ant, DPO w/o Sqrt in circle control etc.). This makes me feel that the DPO algorithm could show unlearning if run for more steps in the future. Do the authors have any intuitions for why this may not be the case? If not, I would recommend experiments to be conducted for longer periods. Especially several methods in the paper in all experiments still seem to be improving/changing and have not reached a stable point in training.
>
> The main difference between DPO and the baselines or ablation algorithms is that DPO has a convergence guarantee. Thus, our intuitive explanation for the difference in performance in the experiments is the theoretical result of convergence. As we discussed in our analysis, in DPO, even if agents update their policies independently, the joint policy will monotonically improve and eventually converge. Therefore, we believe that DPO will not show unlearning if run for more steps.
>
> IDDPG, a version of IQL for continuous action spaces, is still a heuristic algorithm with no convergence guarantee. So it can solve some tasks, but it is not surprising that IQL or IDDPG fails on some tasks. Moreover, the failure of IQL was even observed in a 3x3 matrix game with two players in [1]. The explanation for the failure of IQL is, among others, that IQL cannot coordinate in the exploration [1] and that IQL has the problem of overestimation [2].
>
> DPO w/o Sqrt is an ablation algorithm of DPO in which the term $\sqrt{D_{\operatorname{KL}}}$ is omitted, which means that DPO w/o Sqrt loses the convergence guarantee. As explained in Lemma 3.1, the term $\sqrt{D_{\operatorname{KL}}}$ is a bridge between the joint objective and the independent objective, and if one omits the term $\sqrt{D_{\operatorname{KL}}}$, one loses the guarantee of improvement for the joint policy.
>
>
>
> [1] Claus, Caroline, and Craig Boutilier. "The dynamics of reinforcement learning in cooperative multiagent systems." *AAAI/IAAI* 1998.746-752 (1998): 2.
>
> [2] Tampuu, Ardi, et al. "Multiagent cooperation and competition with deep reinforcement learning." *PloS one* 12.4 (2017): e0172395.
>
> > In the experiments different performances of algorithms in different experiments do not have clear explanations. For instance, why does IQL show instability in MPE, but performance close to the best performing algorithm in SMAC? This may show that the hyper-parameters have not been carefully tuned for these baselines in all experiments.
>
> Thank you for your advice regarding the experiment part. Similar to the discussion above, IQL is a heuristic algorithm with no convergence guarantee, so it is normal for IQL to perform differently in different environments. It has been shown that IQL performs differently even on different tasks in the same environments SMAC [1]. The experiments in [1] show that IQL can achieve near-optimal performance in some maps, while it learns nothing at all in others.
>
> [1] Rashid, Tabish, et al. "Monotonic value function factorisation for deep multi-agent reinforcement learning." *The Journal of Machine Learning Research* 21.1 (2020): 7234-7284.
>
> > In general, I am confused as to what the experiments are trying to accomplish. The paper says that "focus on comparison between DPO and IPPO and just add IQL as baseline". Why are such comparisons helpful? Also, I strongly disagree with the conclusion in the paper "the results show the advantage of DPO over IPPO and IQL, which can be evidence for our theoretical results" -- the algorithm has important differences with respect to the theory and I am not convinced it is acting as an evidence for any theoretical findings.
>
> Our purpose of the experiments is to compare DPO, the algorithm we proposed with convergence guarantee, with the state-of-the-art fully decentralized algorithms IPPO and IQL, which are heuristic and represent the policy-based algorithm and the value-based algorithm, respectively. The results of the experiment show that DPO has better performance and stability on various tasks. Since we set the structure of the neuro-network and the hyper-parameters of DPO and IPPO to be the same, the advantage of DPO lies in the novel policy objective obtained through our theoretical analysis. Although we have admitted the gap between the practical algorithm and theory, our ablation study shows that the two terms $\sqrt{D_{\operatorname{KL}}}$ and $D_{\operatorname{KL}}$ in equation (12) both have an important impact on the performance of DPO. Summarizing the above, we conclude that DPO is effective in the fully decentralized MARL. The reason for the effectiveness is the convergence guarantee obtained in Theorem 3.2.

---

> ### Author Response · Authors · 2023-11-17
> **Response**
>
> > There seems to be some abuse of notations in the theory which makes it hard to follow everything. The objective $J(\pi)$ is not conditioned on the initial state (though it should be conditioned on some initial state). The policy (and joint policy) was never defined. In some cases the Q function and value function are not parameterized by the agent index (Eq for $A_{old}^i$ following Eq.6), and in other cases they are parameterized by the agent index (Eq 7), which makes me more confused.
>
> Thank you for your comment on the notation. In our discussion, the initial state is fixed or obeys a fixed distribution, so we omit the initial state in the objective $J(\pi)$. As for the agent index and the value function, the value function without the agent index is the joint value function related to the joint action, like $A_{old}(s,a_i,a_{-i})$ in the equation you mentioned.The value function with the agent index $i$ is the individual value function of the agent $i$ which is the expectation  of the the joint value function over the policies of the other agents $\pi_{-i}$  from the perspective of agent $i$.  We have added more explanation about the notation in Section 3.1.

---

### Review · Reviewer_PNJm · 2023-10-02

**Summary Of Contributions:**

This paper proposes a fully decentralized policy optimization (DPO) algorithm in cooperative multi-agent reinforcement learning (MARL).
The assumption of the decentralized MARL setting is that each agent observes the state and the joint policy is factorized into a product of local policies of each agent. Motivated by the surrogate loss of the trust-region policy optimization in the single-agent case, the authors devise a surrogate loss of the joint policy that factorizes into a sum of the individual loss of each agent. This surrogate loss involves the square-root of the max-KL divergence and the max-KL-divergence itself. In practice, the agent-wise separable loss is optimized using two adaptive coefficients and the max-KL-divergence is replaced by an average of KL-divergence over roll-out data. Experiments in several cooperative MARL environments show that the proposed DPO can perform better than independent PPO under fully decentralized settings.

**Audience:**

Yes

**Broader Impact Concerns:**

No.

**Claims And Evidence:**

No

**Requested Changes:**

1. Strong assumptions: Using the max-KL divergence seems very stringent. By taking the maximum of $s$, this terms can be very large. This essentially means that, when updating policy, the new policy has to be close to the old policy in each state, which is hard to satisfy when you have a neural network policy. In contrast, in single-agent TRPO/PPO literature, there are already existing works showing that variants of TRPO/PPO archives **global optimality** even when the regularization term is an averaged KL-divergence. E.g., see "Policy Mirror Descent for Reinforcement Learning: Linear Convergence, New Sampling Complexity, and Generalized Problem Classes". I would suggest the authors consider the weaker KL divergence.

2. Unrealistic algorithm: It seems that the algorithm is hard to implement in a decentralized fashion. The reason is that the algorithm needs to solve a policy evaluation subproblem when updating the policies. When implementing the algorithm, essentially we need to let the agents hold their policies fixed until the policy evaluation problem is solved.

3. How do you solve (19) practically?

4. It seems unclear to me why you need both the square-root KL and the KL itself. When the update is small, then squareroot KL is always larger right?

5. Is it possible to use a smaller-scale environment to implement the theoretically verified algorithm in a fully decentralized fashion, and empirically show monotonic improvement?

**Strengths And Weaknesses:**

Strength: The authors develop a new surrogate loss for MARL that guarantees monotonic improvement when the parameters are properly chosen.

Weakness: My major concerns are (i) the theoretical results seem some direct extension from single-agent TRPO, with the help of strong and even unrealistic assumptions, (ii) the algorithm with a theoretical guarantee of monotonic improvement might be hard to implement and the algorithm implemented is different from the theory.

---

> ### Author Response · Authors · 2023-11-17
> **Response**
>
> > Strong assumptions: Using the max-KL divergence seems very stringent. By taking the maximum of $s$, this terms can be very large. This essentially means that, when updating policy, the new policy has to be close to the old policy in each state, which is hard to satisfy when you have a neural network policy. In contrast, in single-agent TRPO/PPO literature, there are already existing works showing that variants of TRPO/PPO archives **global optimality** even when the regularization term is an averaged KL-divergence. E.g., see "Policy Mirror Descent for Reinforcement Learning: Linear Convergence, New Sampling Complexity, and Generalized Problem Classes". I would suggest the authors consider the weaker KL divergence.
>
> Thank you for your advice. Through some derivations, we find that we can use the conclusion in the paper you mentioned or a similar conclusion in [1] to replace the maximum term with the average term in the joint TRPO objective. With some modification to the proof of Lemma 3.2, we can also replace the maximum term with the average term for the inequality in Lemma 3.2. We have updated the details in Section 3.3.
>
> With the modifications above, we can avoid the strong assumption of taking the maximum of $s$ and also make the gap between the theory and the algorithm smaller. Unfortunately, we did not find a way to prove that the joint policy can achieve **global optimality**, since Lemma 3.2 is a lower bound and not an equality. Achieving global optimality in fully decentralized MARL is still an open problem and beyond the scope of our discussion.
>
> [1] Grudzien, Jakub, Christian A. Schroeder De Witt, and Jakob Foerster. "Mirror learning: A unifying framework of policy optimisation." *International Conference on Machine Learning*. PMLR, 2022.
>
> > Unrealistic algorithm: It seems that the algorithm is hard to implement in a decentralized fashion. The reason is that the algorithm needs to solve a policy evaluation subproblem when updating the policies. When implementing the algorithm, essentially we need to let the agents hold their policies fixed until the policy evaluation problem is solved.
>
> The evaluation of the individual critic $Q^{\boldsymbol{\pi}}_i(s,a_i)$ is the unavoidable difficulty in fully decentralized learning, since agents cannot observe the actions or policies of other agents. Therefore, agents have to approximate $Q^{\boldsymbol{\pi}}_i(s,a_i)$ by the samples of rewards, which is unbiased, as our discussion in Section 3.2 shows. The solution for evaluating $Q^{\boldsymbol{\pi}}_i(s,a_i)$ is the on-policy update for the individual critic. Although the on-policy update may be troubled by the trade-off between sample complexity and accuracy, we do not consider it unrealistic. In the practical algorithm, agents do not need to hold their policies fixed until the policy evaluation problem is solved. They only need a certain number of samples under the joint policy $\boldsymbol{\pi}$ and then evaluate their individual critics. IPPO [1] faces the same problem and uses the same solution.
>
> [1] Christian Schroeder de Witt, Tarun Gupta, Denys Makoviichuk, Viktor Makoviychuk, Philip H. S. Torr, Mingfei Sun, and Shimon Whiteson. Is independent learning all you need in the starcraft multi-agent challenge? arXiv preprint arXiv:2011.09533, 2020.
>
> > How do you solve (19) practically?
>
> Practically we replace the objective of IPPO with (19) and keep all other things unchanged. We use gradient descent with Adam optimizer for (19).
>
> > It seems unclear to me why you need both the square-root KL and the KL itself. When the update is small, then squareroot KL is always larger right?
>
> We have discussed this question in Section 3.4. First, we cannot ensure that the KL divergence term is always less than one. Second, the square root term and the KL term have different meanings for the objective function. The square root term from Lemma 3.1 is the bridge between the joint objective and the individual objective. The KL term from TRPO is a bound for the difference between the state distributions of two different policies. Empirically, our ablation studies in Section 4.5 have shown that these two terms have an important impact on the performance of DPO.
>
> > Is it possible to use a smaller-scale environment to implement the theoretically verified algorithm in a fully decentralized fashion, and empirically show monotonic improvement?
>
> We add experiments in a simple two-player matrix game to empirically show monotonic improvement with  the theoretically verified algorithm in Figure 11 in Appendix C.   More detailed discussion is also included in Appendix C.

---

### Review · Reviewer_Cmkb · 2023-11-04

**Summary Of Contributions:**

This paper presents a novel algorithm for cooperative multi-agent reinforcement learning (MARL) in fully decentralized settings, where each agent learns its own policy without communication or parameter sharing. The main contributions and new knowledge are:

1. The authors propose decentralized policy optimization (DPO), a decentralized actor-critic method that guarantees monotonic improvement and convergence of the joint policy by optimizing a novel surrogate objective.
2. The authors derive a decentralized surrogate that can be decomposed into individual objectives for each agent, and show that it is a lower bound of the joint policy improvement.
3. The authors evaluate DPO on various cooperative MARL tasks, covering discrete and continuous action spaces, as well as fully and partially observable environments, and demonstrate that DPO outperforms existing decentralized methods.

**Audience:**

Yes

**Claims And Evidence:**

Yes

**Requested Changes:**

## Critical to securing my recommendation

1. Correct the proof as mentioned in the `Weaknesses` section.
2. Add discussions about recent works that theoretically analyze the monotonic improvement in MAS, such as CoPPO, HAPPO and A2PO.

###Simply strengthen the work

1. Add the performance of IPPO into the ablation for a clear comparison.

**Strengths And Weaknesses:**

## Strengths

This paper analyzes the potential of using trust-region based in decentralized multi-agent reinforcement learning and propose practical algorithms, with abundant ablation studies.

## Weaknesses

Theoretically, there exists **an error** in proving the monotonic improvement bound, which is the **main theoretical contribution** of this paper.
In the proof of the following equation, the authors only prove that the operator $\Gamma_{\pi^{-i}}^{\pi^i}$ is a contraction mapping.
$$
Q_{\pi^{-i}}^{\pi^i}(s, a_i)=E_{\pi^{-i}}[Q^\pi(s, a_i, a_{-i})]
$$

There are two main mistakes:

1. One can not prove that a variable under an operator such as the well-known Bellman operator, converges to a fixed point by only showing the operator is a contraction mapping, since in the algorithm the variable is updated using gradients. Therefore, the authors should consider model learning $Q_{\pi^{-i}}^{\pi^i}(s, a_i)$ as a random process (or random approximation) [1].

2. Even if
$Q_{\pi^{-i}}^{\pi^i}(s, a_i)$ and $E_{\pi^{-i}}[Q^\pi(s, a_i, a_{-i})]$
are accurately proved to converge to the same fixed point, the authors can not directly derive that $E_{\pi^{-i}}[Q^\pi(s, a_i, a_{-i})]$. The two variables are not proved to be equal at all steps. **So the mismatch between the two variables should also be considered when proving the monotonic improvement.**

   [1] Melo F S. Convergence of Q-learning: A simple proof.

---

> ### Author Response · Authors · 2023-11-17
> **Response**
>
> > About the 'error' for the decentralized critic.
>
> In Section 3.2, we only discuss the decentralized critic $Q^{\pi^i}\_{\pi^{-i} }(s,a_i)$ from the perspective of policy evaluation and don't claim any iteration or algorithm like Q-learning.  If we fix the policies of all the other agents $\pi^{-i}$ which become a part of the environment, then the multi-agent problem degenerates to a single-agent problem. From the conclusion in single-agent RL, $Q^{\pi^i}\_{\pi^{-i} }(s,a_i)$ is the Q-function of agent $i$ in this single-agent problem.  We prove that $Q^{\pi^i}\_{\pi^{-i} }(s,a_i)$ and $\mathbb{E}\_{\pi^{-i}}[Q^{{\boldsymbol{\pi}}}(s,a_i,a_{-i})]$ are the same from the perspective of policy evaluation and definition. A similar conclusion with fixed point proof can be found in Lemma 2 in Appendix A.3 of [1] .  Thus, $Q^{\pi^i}\_{\pi^{-i} }(s,a_i)$ and $\mathbb{E}\_{\pi^{-i}}[Q^{{\boldsymbol{\pi}}}(s,a_i,a_{-i})]$ can be seen as the same in our theoretical results.    For your concern of the learning of decentralized critic, we add a proof of the Q-iterations of $Q^{\pi^i}_{\pi^{-i} }(s,a_i)$ in Appendix F.
>
> As for the practical algorithm,  the evaluation of the individual critic $Q^{\pi^i}_{\pi^{-i} }(s,a_i)$ is the unavoidable difficulty in fully decentralized learning.  The solution is the on-policy update for the individual critic. IPPO [2] faces the same problem and uses the same solution. We admit the on-policy update may be troubled by the trade-off between sample complexity and accuracy but it is beyond the scope of our discussion.
>
> [1] Lyu, Xueguang, et al. "Contrasting centralized and decentralized critics in multi-agent reinforcement learning." *arXiv preprint arXiv:2102.04402* (2021).
>
> [2] Christian Schroeder de Witt, Tarun Gupta, Denys Makoviichuk, Viktor Makoviychuk, Philip H. S. Torr, Mingfei Sun, and Shimon Whiteson. Is independent learning all you need in the starcraft multi-agent challenge? arXiv preprint arXiv:2011.09533, 2020.
>
> > Add discussions about recent works that theoretically analyze the monotonic improvement in MAS, such as CoPPO, HAPPO and A2PO.
>
> We have mentioned CoPPO and HAPPO in Section 2 that they are CTDE algorithms and not appropriate for   fully decentralized learning.  HAPPO needs agents to maintain a joint advantage function over the joint action and the agents need the information of the critical function $M^{i_{1:m}}$ from other agents.  These conditions cannot be satisfied in fully decentralized learning.  CoPPO needs the ratios of other agents' policies for policy updates but this information cannot be obtained in fully decentralized learning.  A2PO uses the joint advantage function and agents need the joint policy in the off-policy correction for the calculation of $A^{\boldsymbol{\pi},\hat{\boldsymbol{\pi}}^{i-1}}$. Moreover, A2PO uses the double-clip trick similar to CoPPO which requires the ratios of other agents' policies. Thus, A2PO is also not appropriate for fully decentralized learning. We have added A2PO into the related work. We have included the discussion above into the Appendix D.
>
> > Add the performance of IPPO into the ablation for a clear comparison.
>
> We have updated the figures in ablation study in Section 4.5.

---

### Decision · Action_Editor_PjXX · 2024-01-06

**Recommendation:** Accept as is

**Comment:**

The majority of reviewers were in favor of publication, and the remaining concerns weren't sufficient to override that (e.g. whether the phrasing overly suggested a focus on partially observed domains).

**Audience:**

Multi-Agent policy optimization is a growing subfield, and the proof techniques involved should be accessible to the even broad audience of single agent RL researchers. The experiments also suggest the technique to be scalable, which opens the door to more applied researchers also potentially finding value in the work.

**Claims And Evidence:**

There was some initial confusion about the theoretical justification of this algorithm, but post-revision the theorems seem reasonable. Empirical support straightforward and wasn't significantly flagged by any of the reviewers.